# Nonparametric Distribution Regression Re-calibration

Ádám Jung [1 2]   Domokos M. Kelen [1]   András A. Benczúr [1]

## Abstract

A key challenge in probabilistic regression is ensuring that predictive distributions accurately reflect true empirical uncertainty. Minimizing overall prediction error often encourages models to prioritize informativeness over calibration, producing narrow but overconfident predictions. However, in safety-critical settings, trustworthy uncertainty estimates are often more valuable than narrow intervals. Realizing the problem, several recent works have focused on post-hoc corrections; however, existing methods either rely on weak notions of calibration (such as PIT uniformity) or impose restrictive parametric assumptions on the nature of the error. To address these limitations, we propose a novel nonparametric re-calibration algorithm based on conditional kernel mean embeddings, capable of correcting calibration error without restrictive modeling assumptions. For efficient inference with real-valued targets, we introduce a novel characteristic kernel over distributions that can be evaluated in $\mathcal{O}(n \log n)$ time for empirical distributions of size $n$. We demonstrate that our method consistently outperforms prior re-calibration approaches across a diverse set of regression benchmarks and model classes.

## 1. Introduction

In safety-critical applications, such as autonomous systems or medical domains, the value of a predictive model hinges not only on its raw predictive power but also on its reliability. Models that predict the outcomes of medical procedures or vehicle trajectories must also provide trustworthy estimates of their own certainty. In probabilistic regression, this requires balancing two complementary objectives: concentrating probability mass around the ground truth (*sharpness*) and accurately reflecting the true empirical error distribution (*calibration*). However, common training objectives, such as the negative log-likelihood, tend to reward improvements in the former even when achieved at the expense of the latter. As a result, modern neural networks often fail to achieve calibration, manifesting as narrow predictive intervals that fail to capture the true range of outcomes (Minderer et al., 2021). In high-stakes environments, such miscalibration is dangerous; a system that cannot accurately quantify its own uncertainty provides a false sense of security, potentially leading to severe consequences (Kompa et al., 2021).

Calibration has long been a central problem in both statistical forecasting (Dawid, 1984) and machine learning. Today, the topic continues to attract significant interest through recent works on post-hoc recalibration (Guo et al., 2017; Song et al., 2019; Gruber & Buettner, 2024). The primary tool in this domain is the *Probability Integral Transform* (PIT) (Dawid, 1984; Diebold et al., 1997; Mitchell & Wallis, 2011; Kuleshov et al., 2018), which serves as both a design objective and evaluation criterion by testing if the cumulative distribution function (CDF) values of observed targets are uniformly distributed. However, as also noted by Gneiting & Resin (2023), PIT uniformity is a necessary but insufficient condition: it is a *marginal* property that allows for "error cancellation". Consider an autonomous vehicle: a model that is dangerously overconfident in difficult conditions (e.g., heavy fog) can mask this behavior by being underconfident in easy conditions (e.g., clear weather). As long as errors average out globally, the PIT statistic will appear uniform, concealing the model's unreliability.

The inadequacy of PIT has motivated stronger notions of calibration (Tsyplakov, 2013; Gneiting & Resin, 2023; Widmann et al., 2021; Glaser et al., 2023; Moskvichev & Sejdinovic, 2025); however, existing methods either focus solely on quantifying calibration error or impose restrictive parametric assumptions when attempting to mitigate it. Such constraints limit the use of existing recalibration approaches on complex, real-world data, forcing a choice between flexible but weak methods (PIT-based) or rigorous but unrealistic ones (parametric).

To address this gap, we propose a novel recalibration framework. By leveraging *Conditional Kernel Mean Embeddings* (CKME) to map model representations directly to a cal-

[1]HUN-REN SZTAKI, Budapest, Hungary [2]Eötvös Loránd University, Budapest, Hungary. Correspondence to: Ádám Jung <adam.jung@sztaki.hu>.

*Proceedings of the 43rd International Conference on Machine Learning*, Seoul, South Korea. PMLR 306, 2026. Copyright 2026 by the author(s).

ibrated distribution, our approach avoids both the failure modes of marginal PIT and the restrictiveness of parametric assumptions. Our specific contributions are as follows: First, we introduce a nonparametric recalibration algorithm that enforces the strict property of auto-calibration. Second, to achieve scalability, we propose novel characteristic kernel over distributions, based on the notion of energy distance (Szekely & Rizzo, 2004). For real-valued targets, this kernel can be evaluated in $\mathcal{O}(n \log n)$ time, overcoming the quadratic bottlenecks of standard nonparametric distribution kernels. Third, to better understand the theoretical concepts of this approach, we derive a novel calibration–sharpness decomposition of the population-level error, leveraging a notion of generalized conditional mutual information between the target, feature, and prediction.

In our experiments, we show that models such as Distributional Random Forest (Cevid et al., 2022), Mixture Density Networks (Bishop, 1994), Bayesian Neural Networks (Blundell et al., 2015) are often miscalibrated. We show that our algorithm provides better calibration compared to the parametric re-calibration approach of Song et al. (2019). We evaluate our method by Squared Kernelized Calibration Error (SKCE) (Widmann et al., 2021), a PIT calibration test with Kolmogorov-Smirnov test, and average test set CRPS relative to the same model without recalibration. Our experiments using the UCI Regression Benchmark (Hernandez-Lobato & Adams, 2015) and other data confirm the practical usability of our algorithm and the supporting theory.

## 2. Related Work

Our approach to calibration is grounded in the statistical properties of proper scoring rules. We build on the theories of proper scoring rules (Gneiting & Raftery, 2007), formal notions of calibration (Gneiting & Resin, 2023), and generalized definitions of entropy, divergence, and mutual information induced by scoring rules (Dawid & Musio, 2014).

Statistical tests of calibration have appeared early in statistical literature, with perhaps Dawid (1984) the first to introduce the Probability Integral Transform (PIT) with a goodness-of-fit test. The relation of PIT and calibration is explored among others in (Strähl & Ziegel, 2015; Modeste et al., 2024). Perhaps the first auto-calibration test overcoming the heuristic limitation of the PIT test is introduced in (Tsyplakov, 2013), however it is not consistent in general. In our work, we employ the Squared Kernelized Calibration Error (SKCE) (Widmann et al., 2021) as the most reliable test of calibration.

Towards understanding and decomposing the sources of prediction error, Gruber & Buettner (2024) introduce the notion of the Proper Calibration Error, and provide a simi-

lar decomposition to ours, but without separating aleatoric uncertainty explicitly. They introduce a somewhat limited variance regression recalibration, and provide an algorithm to recalibrate with respect to their notion only.

Most similar to our solution are the recalibration algorithms, out of which we use the best performing ones as baseline. Most important and widely used is PIT recalibration, for which we use the method of Kuleshov et al. (2018). Marx et al. (2023) introduce the trainable calibration metric. During training, they add the kernelized auto-calibration error as regularization to balance between sharpness and calibration. However, with their technique they require a post-hoc recalibration algorithm, for which they use PIT recalibration. While an empirically well performing method, it does not provide any guarantees for auto-calibration.

Closest to our work is (Song et al., 2019), with the key difference that they rely on strong parametric assumptions about the nature of the calibration error. They model the parameters describing the calibration error as a Gaussian process, dependent on the first two moments of the original prediction. Further, they do not report the hypothesis test result of (Widmann et al., 2021), as their publication predates this test. As future work, the authors propose to solve the same problem with a nonparametric approach, which we address in our paper.

Another strongly related result is (Moskvichev & Sejdinovic, 2025), which evaluates calibration with conditional kernel mean embeddings. Just as in their paper, we propose nonparametric calibration. However, our approach differs in the following ways: First, we consider regression, not just classification. Second, we provide a recalibration algorithm instead of merely quantifying the calibration error. Finally, our evaluation includes not only calibration, but also the overall error score.

The calibration of classification models is better developed than that of regression. While Kull et al. (2017; 2019) achieve post-hoc auto-calibration, they rely on parameterizing the calibration map – similarly to Song et al. (2019) – which is restrictive for continuous regression densities. Other methods, such as those by Hebert-Johnson et al. (2018); Luo et al. (2022) rely on the histogram binning of the confidence scores $[0, 1] \subset \mathbb{R}$. Vashistha & Farahi (2025) focus on quantifying a stronger notion of calibration called local calibration error. Finally, Kull & Flach (2015) and Perez-Lebel et al. (2023) introduce and estimate the grouping loss component of the expected error. The notion of grouping loss quantitatively coincides with our information-theoretic definition of lack of sharpness (see Section 4 for details).

Conformal Prediction (CP) is a related framework in that it also utilizes a held-out calibration set to provide / improve

reliability guarantees (Fontana et al., 2023; Chernozhukov et al., 2021; Vovk et al., 2017). However, CP fundamentally differs from our approach in its objectives and outputs. Standard CP aims to construct a prediction interval or region that contains the true target with a user-specified marginal probability. While it is practical for robust, worst-case decision-making, it does not characterize the probability distribution inside the predicted region. Furthermore, the two frameworks operate on different notions of reliability. Standard CP guarantees marginal coverage, and advanced CP methods strive for conditional coverage (conditioned on the feature $X$). In contrast, our proposed framework enforces Auto-calibration (cf. Section 3.3.2), which sits structurally between marginal and full feature-conditional calibration.

## 3. Background

Let $\mathcal{X}$ and $\mathcal{Y}$ be the feature and response spaces, respectively. Let $\mathcal{M}(\mathcal{Y})$ denote the space of probability measures over $\mathcal{Y}$. Given random variables $\xi$ and $\eta$, let $\mathbb{P}_\xi$, $\mathbb{P}_{\xi|\eta}$ and $\mathbb{P}_{\xi,\eta}$ denote the marginal distribution of $\xi$, the conditional distribution of $\xi$ given $\eta$ and the joint distribution of $(\xi, \eta)$, respectively. Suppose that we have an i.i.d. sample $\mathcal{D} = \{(x_i, y_i)\}_{i=1}^n$ from the joint distribution of the feature and the target $(X, Y) \sim \mathbb{P}_{X,Y}$.

### 3.1. Proper Scoring Rules

Distribution fitting can be performed under various notions of alignment of the observations and predicted distributions. We will consider the concept of *Scoring Rules* (Gneiting & Raftery, 2007), as it provides a unifying framework.

Let $S : \mathcal{M}(\mathcal{Y}) \times \mathcal{Y} \to \mathbb{R}$ be a function that assigns a score $S(q, y)$ to a prediction $q \in \mathcal{M}(\mathcal{Y})$ and an observation $y \in \mathcal{Y}$. We will use negatively oriented scores, i.e., smaller scores are better. $S$ is said to be *strictly proper* if

$$\mathbb{E}[S(q, Y)] > \mathbb{E}[S(\mathbb{P}_Y, Y)] \ ,$$

for all $q \in \mathcal{M}(\mathcal{Y}) \setminus \{\mathbb{P}_Y\}$. That is, in expectation the minimum score is uniquely obtained at the true distribution of the target.

The well known negative log-likelihood metric corresponds to the *logarithmic score* $S(q, y) = -\log p_q(y)$ (with $p_q$ denoting the probability density function of $q$), and is a strictly proper scoring rule.

For real-valued distributions which cannot be represented as a density, the *Continuous Ranked Probability Score* (CRPS) is a widely used (strictly proper) scoring rule:

$$S(q, y) = \mathbb{E}|M - y| - \tfrac{1}{2}\mathbb{E}|M - M'| \ , \qquad (1)$$

where $M \sim q$ and $M'$ is an i.i.d. copy of $M$.

The excess score stemming from making an imperfect prediction is called the *divergence* and is denoted with

$$d(q^*, q) = \mathbb{E}_{M \sim q^*}[S(q, M)] - \mathbb{E}_{M \sim q^*}[S(q^*, M)] \ . \quad (2)$$

The divergence is always non-negative, and for strictly proper scoring rules $d(q^*, q) = 0$ implies $q = q^*$. The second term in (2) is the expected score of a perfect prediction, which is called the *generalized entropy* and denoted with

$$H(q) = \mathbb{E}_{M \sim q}[S(q, M)] \ . \qquad (3)$$

For our demonstration purposes, we will introduce a *conditional* version of the generalized notion of mutual information (Dawid & Musio, 2014). It quantifies conditional dependence via the expected reduction of entropy, as follows.

**Definition 3.1.** Let $Y, X, Z$ be jointly distributed random variables. The *generalized conditional mutual information* of $Y$ and $X$ given $Z$, induced by the entropy function $H$, is defined as

$$I(Y; X|Z) = \mathbb{E}_{Z,X}\left[H(\mathbb{P}_{Y|Z}) - H(\mathbb{P}_{Y|Z,X})\right] \ .$$

If $H$ is induced by a strictly proper scoring rule, then $I(Y; X|Z)$ is nonnegative and $I(Y; X|Z) = 0$ iff $Y$ is conditionally independent of $X$ given $Z$. See Appendix A.3 for details. For the logarithmic score, $H$ and $d$ coincides with the differential entropy, and Kullback-Leibler divergence, respectively. Whereas in the case of the CRPS score, $H$ is half the mean absolute error

$$H(q) = \tfrac{1}{2}\mathbb{E}|M - M'| \ ,$$

with $M, M' \sim q$ i.i.d., and $d(q^*, q)$ equals to

$$\mathbb{E}|M - M^*| - \tfrac{1}{2}\mathbb{E}|M - M'| - \tfrac{1}{2}\mathbb{E}|M^* - M^{*\prime}| \ , \quad (4)$$

where $M^*, M^{*\prime} \sim q^*$ i.i.d., independent of $M$ and $M'$. Equation (4) is a well known metric in the statistical literature, called the energy distance (Szekely & Rizzo, 2004; Baringhaus & Franz, 2004).

### 3.2. Kernel Mean Embedding of Distributions

Kernel based algorithms are a powerful and well-developed branch of statistical machine learning. We will present only the most important concepts needed to introduce kernel mean embeddings, which is a nonparametric technique capable of estimating distances between distributions efficiently, and estimate conditional distributions, applicable over very general feature and target spaces. See Muandet et al. (2017) for a gentle introduction.

Consider a set $\mathcal{X}$ and a positive definite kernel $k : \mathcal{X} \times \mathcal{X} \to \mathbb{R}$. It is a well known result of Aronszajn (1950) that every

positive definite kernel uniquely defines a *Reproducing Kernel Hilbert Space* $(\mathcal{H}, \langle \cdot, \cdot \rangle_{\mathcal{H}})$ and vice versa. Note that $\mathcal{H}$ is a Hilbert space of functions $\mathcal{X} \to \mathbb{R}$. For every $x \in \mathcal{X}$ the *canonical feature map* $\phi(x) := k(x, \cdot)$ is an element of $\mathcal{H}$.

Consider a random variable $X \sim \mathbb{P}_X$, taking values in $\mathcal{X}$. The expected value

$$\mu_X = \mathbb{E}\left[\phi(X)\right] \tag{5}$$

is called the *kernel mean embedding* (KME) of $\mathbb{P}_X$. If the kernel $k$ is so-called *characteristic*, then the feature map $\phi$ is rich enough, so that the expected value (5) encodes the whole distribution $\mathbb{P}_X$, i.e. the mapping $\mathbb{P}_X \mapsto \mu_X$ is injective.

### 3.2.1. DISTANCE OF MEAN EMBEDDINGS

The norm of $\mathcal{H}$ (induced by $\langle \cdot, \cdot \rangle_{\mathcal{H}}$) is a powerful tool to define a distance on distributions, via their kernel mean embeddings. Given another random variable $Y \sim \mathbb{P}_Y$ independent of $X$, it is a well known fact that

$$\|\mu_X - \mu_Y\|_{\mathcal{H}}^2 = \mathbb{E}\left[k(X, X')\right] + \mathbb{E}\left[k(Y, Y')\right] \\ - 2\mathbb{E}\left[k(X, Y)\right] , \tag{6}$$

where $X'$ and $Y'$ are i.i.d. copies of $X$ and $Y$, respectively.

For empirical distributions $\hat{\mathbb{P}}_X = \frac{1}{n}\sum_{i=1}^{n}\delta_{x_i}$ and $\hat{\mathbb{P}}_Y = \frac{1}{m}\sum_{i=1}^{m}\delta_{y_i}$, it is easy to see that the distance of their empirical kernel mean embeddings $\hat{\mu}_X = \frac{1}{n}\sum_{i=1}^{n}\phi(x_i)$ and $\hat{\mu}_Y = \frac{1}{m}\sum_{i=1}^{m}\phi(y_i)$ can be evaluated in $\mathcal{O}((n+m)^2)$ time, via Equation (6).

An interesting fact is that in the special case of $\mathcal{X} = \mathbb{R}$, it is possible to reduce this time complexity to $\mathcal{O}(n \log n + m \log m)$, when the kernel $k$ is either the Laplace kernel $k(u, v) = e^{-|u-v|/\sigma}$ or the Brownian motion covariance kernel $k(u, v) = |u| + |v| - |u - v|$. See (Bodenham & Kawahara, 2023; Baringhaus & Franz, 2004; Sejdinovic et al., 2013) for details.

### 3.2.2. CONDITIONAL KERNEL MEAN EMBEDDING

Let $l : \mathcal{Y} \times \mathcal{Y} \to \mathbb{R}$ be a kernel on the target space, with canonical feature map $\psi(y) := l(y, \cdot)$. Since we are interested in estimating certain conditional distributions of the target, a central object of this work will be the *Conditional Kernel Mean Embedding* (CKME), defined as

$$\mu_{Y|X} = \mathbb{E}\left[\psi(Y) \mid X\right] .$$

The estimation of $\mu_{Y|X=x}$ based on an i.i.d. sample $\{(y_i, x_i)\}_{i=1}^{n}$ from $\mathbb{P}_{Y,X}$ and a query point $x \in \mathcal{X}$ can be done as

$$\hat{\mu}_{Y|X=x} = \sum_{i=1}^{n}\psi(y_i)\beta_i(x) ,$$

where $\beta(x) = (K + \lambda I_n)^{-1}(k(x, x_1), \ldots, k(x, x_n))^T$ with $\lambda > 0$ being a regularization parameter, $K$ the kernel matrix $[k(x_i, x_j)]_{i,j=1}^{n}$ and $I_n \in \mathbb{R}^{n \times n}$ the identity matrix. See (Song et al., 2009; Park & Muandet, 2020) for details.

## 3.3. Notions of Calibration

### 3.3.1. PIT CALIBRATION

A basic notion of calibration in regression is defined via the *Probability Integral Transform* (PIT) of the predictions:

$$Z := F_Q(Y) ,$$

where $F_Q$ is the cumulative distribution function (CDF) of the predicted distribution $Q$.

If $Z \sim U[0, 1]$, then we say that the model is *PIT calibrated*, which implies that the predicted quantiles match the empirical frequencies of observing the target below the given quantile.

The reason why PIT calibration is a weak notion of calibration is that it can easily happen that model errors cancel out on average (e.g. systematic over- and under-estimation of the target), leading to $Z \sim U[0, 1]$. Therefore, it is possible to satisfy this notion of calibration with unreliable uncertainty estimates. Another problem is that the reliance on CDFs restrict the applicability of this notion to the real-valued target setting. See (Gneiting & Resin, 2023) for interesting negative examples.

### 3.3.2. AUTO-CALIBRATION

A much stronger notion of calibration can be motivated by the idea of enforcing PIT calibration conditionally on the predictions, i.e. requiring $Z|Q \sim U[0, 1]$. This notion is called *auto-calibration* or *calibration in the strong sense*, and is defined more generally (Tsyplakov, 2013) via the condition

$$Q = \mathbb{P}_{Y|Q} .$$

Auto-calibration implies that the predicted distribution $Q$ matches the true conditional distribution of the target given the prediction itself. See (Gneiting & Resin, 2023) for a detailed discussion of the hierarchies between different notions of calibration. In particular PIT and other weak notions of calibration follows from auto-calibration under mild technical assumptions.

### 3.3.3. HYPOTHESIS TESTING CALIBRATION

Testing PIT calibration can be straightforwardly done via goodness-of-fit tests (e.g. Kolmogorov-Smirnov test), applied to the PIT values computed on a test data split.

Testing auto-calibration needs more sophisticated approaches. An early attempt was made by Tsyplakov (2013),

who proposed a test based on checking the correlation of certain real-valued transformations (such as mean or a given quantile) of the predictions and the PIT values. However, this test is not consistent in general, and can only be applied on real-valued targets.

A consistent and very generally applicable hypothesis test was introduced by Widmann et al. (2021), based on the *Squared Kernelized Calibration Error* (SKCE), which is defined as the (squared) distance of the mean embedding of the joint distribution of $(Q, Y)$ and the joint distribution of $(Q, M)$, where $M \sim Q$, i.e.

$$\text{SKCE} = \|\mu_{Q,M} - \mu_{Q,Y}\|_{\mathcal{H}}^2 \ .$$

Here $\mathcal{H}$ corresponds to a kernel $k$, which is defined on the product space $\mathcal{M}(\mathcal{Y}) \times \mathcal{Y}$. Usually $k$ is constructed as a product kernel $k((q, y), (q', y')) = k_1(q, q')k_2(y, y')$, where $k_1$ and $k_2$ are kernels over $\mathcal{M}(\mathcal{Y})$ and $\mathcal{Y}$ respectively.

Note that if $k$ is characteristic on $\mathcal{M}(\mathcal{Y}) \times \mathcal{Y}$, then SKCE $= 0$ implies

$$\mathbb{P}_{Q,M} = \mathbb{P}_{Q,Y} \ ,$$

which further implies auto-calibration. See (Widmann et al., 2021) for technical details and (Glaser et al., 2023) for a more efficient variant, applicable to unnormalized densities.

We argue that the assessment of a re-calibration algorithm must rely on hypothesis testing of the resulting predictions using an appropriate statistical test. In particular, Figures 1 and 3 present the evaluation results for the SKCE auto-calibration test and the Kolmogorov-Smirnov-based PIT calibration test, respectively.

## 4. Calibration vs. Sharpness Principle

There are usually two distinguished sources of uncertainty in probabilistic modeling. The first is called *aleatoric* uncertainty, which stems from the inherent randomness of the target given the features. It cannot be reduced, unless one introduces new features that describe more information about the target. The other source of uncertainty is referred to as *epistemic* uncertainty, which is the result of insufficient training data, and is a lack of knowledge which can be entirely eliminated in the limit of $n \to \infty$.

The paradigm of maximizing *sharpness* subject to *calibration* was first introduced by Gneiting et al. (2007). Calibration corresponds to how accurately the model represents both aleatoric and epistemic uncertainty, whereas sharpness measures the informativeness of the predictions, i.e., the extent to which the predictions capture the information provided by the features about the target.

The performance of a model $Q$ is usually quantified via the expected error score $\mathbb{E}[S(Q, Y)]$ (such as the negative log-likelihood) it achieves on the whole data distribution. This

however conflates the two fundamentally different sources of error: *i)* calibration error, and *ii)* lack of sharpness. Consequently, a model with low overall score may not be calibrated, i.e., reliable.

To present this argument more formally, we state the following lemma.

**Lemma 4.1.** *The sum of calibration error and lack of sharpness is equal to the divergence from perfect predictions, i.e., the expected error score $\mathbb{E}[S(Q, Y)]$ is equal to*

$$\underbrace{\mathbb{E}\left[d\left(\mathbb{P}_{Y|Q}, Q\right)\right]}_{\text{calibration error}} + \underbrace{I(Y; X|Q)}_{\text{lack of sharpness}} + \underbrace{\mathbb{E}\left[H(\mathbb{P}_{Y|X})\right]}_{\text{aleatoric uncertainty}} \ . \quad (7)$$

*Proof.* See Appendix A.1. □

Here the *aleatoric uncertainty* term has nothing to do with the model, it just captures the irreducible inherent randomness of the target, given the features.

We define *lack of sharpness* as the conditional mutual information (Definition 3.1) between the target and the feature given the prediction. That is, the amount of information the feature carries about the target, beyond what is already captured by the prediction. For a perfectly sharp model $I(Y; X|Q) = 0$, i.e., the feature and the target are conditionally independent given the prediction.

Lack of sharpness[1] quantifies the excess entropy of the *target* that the model does not even attempt to capture, even though it could, in principle, be modeled from the features. A sharp model is often associated with low entropy predictions (e.g. narrow confidence intervals in the real-valued setting). In this formalism, however, this is only a consequence, not the primary definition of sharpness. Narrower confidence intervals arise only from the combination of increased sharpness and accurate uncertainty representation, i.e., low calibration error. Consequently, sharpness only *enables* predictions to be more certain.

Our mutual information based notion of sharpness coincides with the so-called *grouping loss* $\mathbb{E}\left[d\left(\mathbb{P}_{Y|X}, \mathbb{P}_{Y|Q}\right)\right]$ introduced by Kull & Flach (2015). See Appendix B for a direct comparison, where we also establish a formal presentation of the sharpness calibration paradigm conjectured by Gneiting et al. (2007).

By *calibration error* quantified by the first term of (7), we refer to auto-calibration, which is calibration in the strong sense of Section 3.3.2. In other words, when the features are assumed to be *hidden*, the realization of the target corresponding to a *given prediction* is distributed identically to a synthetic sample drawn from that prediction, reflecting our

---

[1]Note that $0 \le I(Y; X|Q) \le I(I; X)$. Because $I(Y; X|Q)$ quantifies the *lack* of sharpness, it is a negatively oriented metric; thus, a lower value indicates a sharper model.

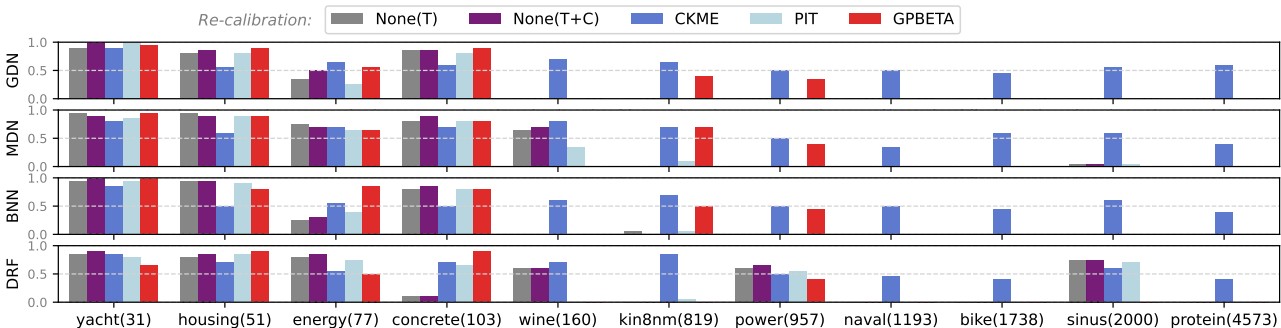

*Figure 1.* Fraction of random train-test splits where the hypothesis of auto-calibration was accepted by SKCE at $\alpha = 5\%$. The numbers after the dataset name indicate the size of the test set $|\mathcal{D}_{\text{test}}|$, allowing the power of the hypothesis test to be assessed. See Appendix H.2 for detailed results.

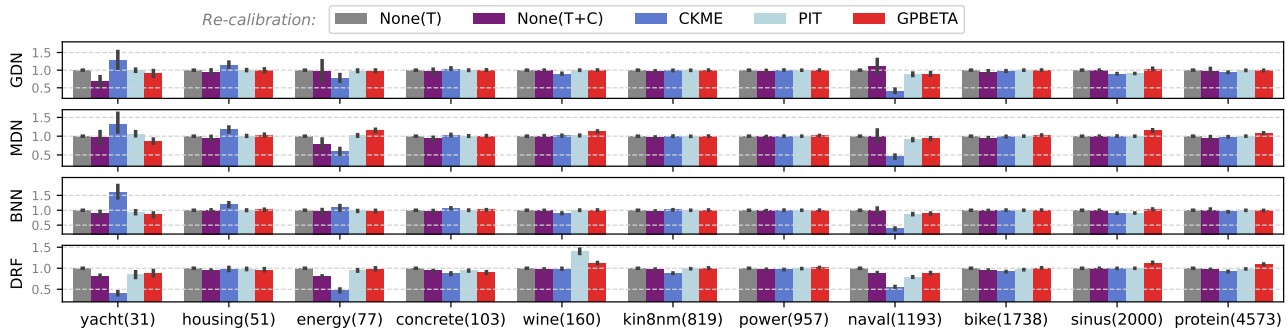

*Figure 2.* CRPS loss relative to the base model trained only on the train set ($\mathrm{None(T)}$). See Appendix H.2 for detailed results. Note that satisfying calibration is trivial if one does not care about sharpness (e.g., by always predicting the marginal distribution of the target $\mathbb{P}_Y$). Since estimating sharpness itself is prohibitively challenging, requiring access to $\mathbb{P}_{Y|X}$, we argue that the best way to assess a re-calibration method is to simultaneously evaluate its calibration performance (Fig. (1)) and the resulting expected score of the model (this Figure).

general expectation on the behavior of a reliable uncertainty estimate.

The first two terms of Equation (7) are nonnegative by definition, and are both $0$ for the perfect model $Q = \mathbb{P}_{Y|X}$. The divergence from perfect predictions, i.e., $\mathbb{E}\left[d\left(\mathbb{P}_{Y|X}, Q\right)\right]$ can be manifested in arbitrary combinations of the first two terms. It follows that the same expected score may result from either a well-calibrated model or a sharp yet unreliable one.

## 5. Re-calibration

Motivated by the calibration–sharpness principle presented in Section 4, in this section, we give a nonparametric, kernel based algorithm to recalibrate a given model while preserving its sharpness, thereby obtaining reliable and useful predictions. To formalize the goal of correcting calibration

error, we introduce the following definition.

**Definition 5.1.** Assume we have a prediction, target, feature tuple $(Q, Y, X) \sim \mathbb{P}_{Q,Y,X}$. Let us define the *recalibrated prediction* $\widetilde{Q}$ as

$$\widetilde{Q} = \mathbb{P}_{Y|Q} \ .$$

The following proposition shows that $\widetilde{Q}$ indeed achieves our objective. See (Bröcker, 2009, Appendix A) and (Widmann, 2021) for a similar proposition stated for the classification setting.

**Proposition 5.2.** *The recalibrated prediction $\widetilde{Q}$ is Auto-calibrated, and has the same sharpness as the original prediction $Q$, i.e., we have*

$$\widetilde{Q} = \mathbb{P}_{Y|\widetilde{Q}} \quad and \quad I(Y; X|\widetilde{Q}) = I(Y; X|Q) \ . \quad (8)$$

*Proof.* See Appendix A.2. □

## 5.1. Non-parametric Calibration Map Estimation

Having established the desired properties of $\widetilde{Q}$, the remaining challenge is to estimate the *calibration map* $Q \mapsto \widetilde{Q}$. In the pioneering work of Song et al. (2019), the authors assumed that $Q$ and $\widetilde{Q}$ are close enough that their difference can be described by a low-dimensional parametric transformation, where the parameters are dependent on the first two moments of $Q$.

We relax the heuristic and restrictive assumption on $Q$ and $\widetilde{Q}$, and estimate the calibration map in a fully nonparametric manner using conditional kernel mean embeddings (Song et al., 2009; Park & Muandet, 2020).

In the first step, we embed the predictions of the original model into an RKHS over distributions, i.e., one induced by a kernel $k : \mathcal{M}(\mathcal{Y}) \times \mathcal{M}(\mathcal{Y}) \to \mathbb{R}$. A general recipe for this is given by the so called *Gaussian-type kernels* introduced by Christmann & Steinwart (2010), namely

$$k(q, q') = \exp\left(-\sigma^2 \left\| \mu_M - \mu_{M'} \right\|_{\mathcal{H}_r}^2\right) . \quad (9)$$

Here $\mu_M$ and $\mu_{M'}$ are the kernel mean embeddings of $q$ and $q'$, respectively, in a RKHS $\mathcal{H}_r$ over $\mathcal{Y}$, and $\sigma$ is a bandwidth parameter. The induced kernel $k$ is characteristic provided that $r$ is characteristic.

*Remark* 5.3. The assumptions regarding the original feature space $\mathcal{X}$ and the joint distribution of $(X, Y)$ are treated implicitly in this work, as they are entirely determined by the capabilities of the chosen base model. In contrast, nonparametric re-calibration imposes requirements primarily on the target space $\mathcal{Y}$. Specifically, $\mathcal{Y}$ must be a *compact metric space* to admit a universal Gaussian-type kernel (Christmann & Steinwart, 2010). Furthermore, no restrictions are placed on the joint distribution of $(Q, Y)$, as conditional kernel mean embeddings are universally consistent estimators (Park & Muandet, 2020).

Given the Gaussian-type kernel $k$ and a kernel $l : \mathcal{Y} \times \mathcal{Y} \to \mathbb{R}$ that encodes the targets via its canonical feature map $\psi : \mathcal{Y} \to \mathcal{H}_l$, the CKME framework yields an estimator of $\widetilde{Q}$ given $Q$ of the form

$$\hat{\mu}_{\widetilde{Q}} = \sum_{i=1}^{n} \psi(y_i) \beta_i(Q) .$$

The coefficient vector $\beta(Q) \in \mathbb{R}^n$ is computed as

$$\beta(Q) = (K + \lambda I_n)^{-1} (k(Q, q_1), \ldots, k(Q, q_n))^T , \quad (10)$$

where $K \in \mathbb{R}^{n \times n}$ is the Gram matrix $[k(q_i, q_j)]_{i,j=1}^n$, $\lambda > 0$ is a regularization parameter, and $\{(q_i, y_i)\}_{i=1}^n$ is a size $n$ calibration data-set, containing predictions $q_i$ of the original model when we observed $(X, Y) = (x_i, y_i)$.

Although $\hat{\mu}_{\widetilde{Q}}$ is a consistent estimator of $\mu_{\widetilde{Q}}$ (see Park & Muandet, 2020, Theorem 4.4.), note that we only obtain

---

**Algorithm 1** Pipeline of Post-hoc Re-calibration

**Input:** Datasets $\mathcal{D}_{\text{train}} = \{(x_i, y_i)\}_{i=1}^{n_{\text{train}}}$ and $\mathcal{D}_{\text{cal}} = \{(x_j, y_j)\}_{j=1}^{n_{\text{cal}}}$, query point $x^* \in \mathcal{X}$
Train an $f : \mathcal{X} \to \mathcal{M}(\mathcal{Y})$ model on $\mathcal{D}_{\text{train}}$
Make predictions $\{q_j := f(x_j)\}_{j=1}^{n_{\text{cal}}}$ on $\mathcal{D}_{\text{cal}}$ using $f$
Make a prediction $q^* := f(x^*)$ for the query point
Use Algorithm 2 on $\{(q_j, y_j)\}_{j=1}^{n_{\text{cal}}}$ and $q^*$ to get $\tilde{q}^*$
**Output:** Re-calibrated prediction $\tilde{q}^*$

---

**Algorithm 2** Perform Nonparametric Re-calibration

**Input:** Dataset $\{(q_j, y_j)\}_{j=1}^{n_{\text{cal}}}$, query prediction $q^* \in \mathcal{M}(\mathcal{Y})$
Build kernel matrix $K = [k(q_i, q_j)]_{i,j=1}^{n_{\text{cal}}}$ via Eq. (9)
Compute coefficient $\beta(q^*) \in \mathbb{R}^{n_{\text{cal}}}$ via Eq. (10)
Let $w$ be the projection of $\beta(q^*)$ to $\Delta_{n_{\text{cal}}}$
**Output:** Re-calibrated prediction $\tilde{q}^* := \sum_{j=1}^{n_{\text{cal}}} w_j \delta_{y_j}$

---

the kernel mean embedding of the recalibrated prediction, rather than an explicit representation of the distribution itself. Recovering an estimate $\hat{\widetilde{Q}}$ corresponds to the distributional pre-image problem (Muandet et al., 2017, sec. 3.8.1). In our approach, this is solved by projecting the weight vector $\beta(Q)$ onto the probability simplex $\Delta_n$ to obtain an empirical distribution of the form

$$\hat{\widetilde{Q}} = \sum_{i=1}^{n} w_i \delta_{y_i} , \quad w \in \Delta_n .$$

See Appendix C for additional notes on the distribution pre-image problem. Algorithms 1 and 2 provide an overview of the proposed nonparametric re-calibration method.

## 5.2. The Energy Distance Kernel (EDK)

Up to this point, we have not made any assumptions on the target space $\mathcal{Y}$; in particular, we have not restricted ourselves to recalibrating real-valued distributions, as is done in CDF-based approaches such as (Kuleshov et al., 2018; Song et al., 2019). However, in order to obtain a more efficient algorithm in the special case $\mathcal{Y} = \mathbb{R}$, we propose the *Energy Distance Kernel* (EDK): a specific instantiation of the Gaussian-type kernel (9) by the choice

$$r(u, v) = |u| + |v| - |u - v| .$$

In this case, $\left\| \mu_M - \mu_{M'} \right\|_{\mathcal{H}_r}^2$ coincides with the so-called energy distance (cf. Eq. (4); (Szekely & Rizzo, 2004; Sejdinovic et al., 2013)), which admits closed-form expressions for many well-known parametric distribution families and, more importantly, can be evaluated for empirical distributions of size $m$ in $\mathcal{O}(m \log m)$ time. This contrasts with the quadratic complexity (cf. Eq. (6)) incurred when using

an arbitrary kernel $r$ on $\mathcal{Y}$. Efficiency is crucial in practice, since constructing the kernel matrix $K$ requires $\mathcal{O}(n^2)$ evaluations of $k$.

# 6. Experiments

We perform a comprehensive benchmark of our proposed CKME based recalibration algorithm described in Section 5. We compare it against the recalibration methods of (Kuleshov et al., 2018) (PIT) and (Song et al., 2019) (GPBETA), as well as against uncalibrated original models trained on $\mathcal{D}_{\text{train}}$ (None(T)) or on $\mathcal{D}_{\text{train}} \cup \mathcal{D}_{\text{cal}}$ (None(T + C)), to ensure fair comparisons. We report $p$-values from auto-calibration and PIT-calibration hypothesis tests (cf. Section 3.3.3; see aggregated results on Figure 1 and 3 respectively), as well as the mean CRPS score (1) achieved on the test set, across several real-world datasets and a range of machine learning models to be recalibrated.

We provide an empirical comparison with a conformal prediction method in Appendix D. An ablation study on the efficiency of the EDK, a wall-clock time comparison against PIT recalibration, and a direct comparison of our recalibration framework against standard kernelized distribution regression can be found in Appendix E, G and F, respectively.

Our experiment code is publicly available at `https://github.com/adamgnuj/recalibration_icml2026`.

## 6.1. Datasets

We use the UCI regression benchmark datasets (Hernandez-Lobato & Adams, 2015), which consist of nine real-world datasets with 20 predefined train-validation-test splits (10% test size, with 20% of the training set held out for validation). The exact splits were taken from the repository of Gal & Ghahramani (2016).

In addition, we evaluate our algorithm on the Bike Sharing dataset introduced by Fanaee-T & Gama (2014), as well as on a synthetic data set with bimodal target distribution (see Appendix H.1 for details).

## 6.2. Base Models

We evaluate our recalibration method on a diverse set of probabilistic regression models. Specifically, we consider Distributional Random Forests (DRF; Cevid et al., 2022), Mixture Density Networks (MDN; Bishop, 1994) using the implementation of Kelen et al. (2025), and Bayesian Neural Network–based MDNs (BNN; Blundell et al., 2015), also following the implementation of Kelen et al. (2025). In addition, we include a single-component Mixture Density Network, corresponding to a heteroscedastic Gaussian density network (GDN), as a simpler baseline model.

## 6.3. Experiment Setup

We treat the validation split of the original dataset as a calibration set, denoted by $\mathcal{D}_{\text{cal}}$. For every combination of dataset, base model, and recalibration method, we evaluate performance on the test set $\mathcal{D}_{\text{test}}$, collecting results over 20 repetitions of model training, recalibration (when applicable), and testing. When necessary, 20% of the data available for training is held out as a validation set.

Note that in the case of (GPBETA; Song et al., 2019), the official implementation[2] can only operate if the output of the base model is a single Gaussian. Consequently, we benchmarked base models with GPBETA recalibration only after approximating the base model's predicted distribution using a Gaussian fitted to its first two moments.

The hypothesis test of Widmann et al. (2021) was performed using the authors' implementation,[3] with the Energy Distance Kernel used as the kernel over distributions (c.f. Section 5.2). The kernel $l$ over the target was set to the Laplace kernel, and all kernel bandwidth parameters were chosen using the median heuristic. The regularization parameter $\lambda$ was numerically optimized using a 5-fold cross validation approach on the calibration set. The loss of CKME regression was used as the objective function, i.e., the RKHS ridge regression objective that minimizes the regularized squared distance between the canonical feature maps of the targets and the estimated conditional mean operator.

## 6.4. Discussion

The aggregated results in Figure 1 demonstrate that, with the exception of our proposed nonparametric recalibration approach, there was generally sufficient evidence to reject the hypothesis of auto-calibration across most datasets by the SKCE test (excluding those with extremely small sample sizes). This highlights the effectiveness of our approach to correct calibration error superior to previous attempts.

Regarding PIT calibration, Figure 3 indicates that the method of Kuleshov et al. (2018) remains the most effective. Still, our approach achieves performance comparable to the original models and remains more effective than GPBETA, while simultaneously addressing the stronger notions of calibration discussed previously.

While sharpness cannot be assessed directly, Figure 2 and Table 1 show that our recalibration method usually was able to marginally improve on the overall score compared to the base model it modified. This suggests that even if there might be some loss of sharpness, it is less important given the improvement on calibration.

---

[2] `https://github.com/Srceh/DistCal`
[3] `https://github.com/devmotion/CalibrationTests.jl`

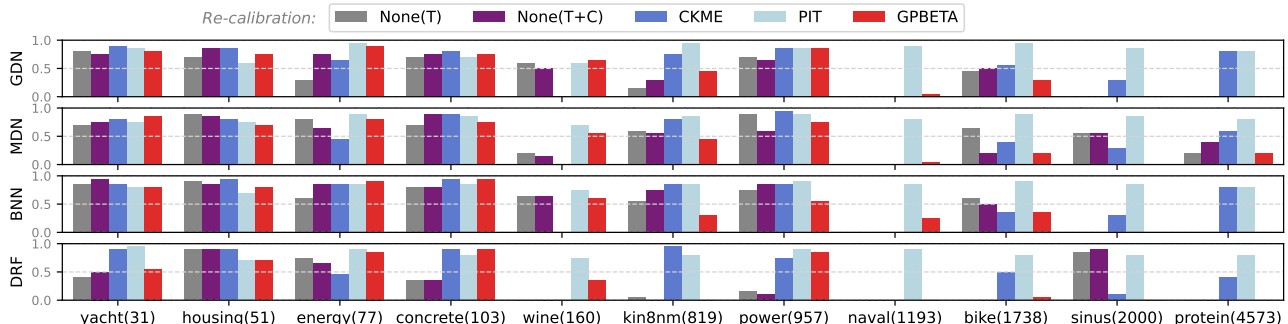

*Figure 3.* Ratio of splits when the hypothesis of PIT-calibration was accepted at $\alpha = 5\%$. The numbers after the dataset name indicate the size of the test set $|\mathcal{D}_{\text{test}}|$, allowing the power of the hypothesis test to be assessed. See Appendix H.2 for detailed results.

By examining Table 1 carefully, we observe that in addition to our algorithm, the base model trained on $\mathcal{D}_{\text{train}} \cup \mathcal{D}_{\text{cal}}$ provides comparable best scores. The outcomes are consistent with the calibration–sharpness principle discussed in Section 4: data can be utilized to either improve calibration or enhance the overall predictive score. Importantly, we demonstrate the motivating negative example for our work: a model can improve its overall score while remaining significantly miscalibrated. This paradox emphasizes the need for careful testing and correction of calibration alongside standard performance metrics.

## 7. Limitations

Our method, like many nonparametric kernel-based approaches, has inherent limitations. Specifically, measuring the distance between complex distributions is fundamentally challenging and can be computationally intensive when there is no explicit representation of the predictions that is easy to work with. Standard kernel methods generally scale with $\mathcal{O}(n_{\text{cal}}^3)$ time complexity due to solving the linear system in Eq. (10).

The targeted notion of calibration may also be too weak for specific applications that require strict local (feature-conditional) calibration (see, e.g., Luo et al., 2022). Furthermore, in contrast to the method of Song et al. (2019), which outputs a calibrated probability density function (PDF), our approach inherently yields an empirical distribution. Consequently, even if the base model predicts continuous densities, our method discards its density estimation structure.

## 8. Conclusions

In this work, we examined the limitations of commonly used calibration techniques in safety-critical regression settings and demonstrated that predictive accuracy alone is insuffi-

cient to guarantee reliable uncertainty estimates. In particular, we highlighted the limitations of PIT-based calibration. To address this gap, we introduced a novel recalibration framework based on Conditional Kernel Mean Embeddings (CKME), which directly maps model representations to calibrated predictive distributions without relying on restrictive parametric assumptions.

Empirical results on the UCI Regression Benchmark and additional datasets show that widely used models are frequently miscalibrated, even when they achieve strong predictive scores. Across these experiments, our method consistently improves calibration relative to state-of-the-art recalibration methods, validating both the theoretical foundations and the practical utility of the proposed framework. The results also reinforce the calibration–sharpness trade-off: available data can be used either to improve calibration or to enhance predictive performance, but gains in one do not necessarily imply gains in the other. Crucially, we demonstrated a negative example in which a model achieves a better overall score while remaining significantly miscalibrated, underscoring the danger of relying solely on standard performance metrics. Together, these findings emphasize the necessity of explicitly testing and correcting calibration, particularly in high-stakes applications where reliable uncertainty quantification is as important as pointwise predictive accuracy.

## Impact Statement

This paper presents work whose goal is to advance the field of machine learning. There are many potential societal consequences of our work, none of which we feel must be specifically highlighted here.

## Acknowledgements

Support from PROACTIF CHIPS Joint Undertaking (JU) under Grant Agreement No. 101194239.

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

# A. Proofs

## A.1. Proof of Lemma 4.1

*Proof.* Using the law of iterated expectations, we have

$$\mathbb{E}\left[S(Q,Y)\right] = \mathbb{E}_Q\left[\mathbb{E}_{Y|Q}\left[S(Q,Y)\mid Q\right]\right] \tag{11}$$

$$= \mathbb{E}_Q\left[\mathbb{E}_{Y|Q}\left[S(Q,Y) - S(\mathbb{P}_{Y|Q},Y) + S(\mathbb{P}_{Y|Q},Y)\mid Q\right]\right] \tag{12}$$

$$= \mathbb{E}_Q\left[\mathbb{E}_{Y|Q}\left[S(Q,Y) - S(\mathbb{P}_{Y|Q},Y)\mid Q\right]\right] + \mathbb{E}_Q\left[\mathbb{E}_{Y|Q}\left[S(\mathbb{P}_{Y|Q},Y)\mid Q\right]\right] \tag{13}$$

$$= \mathbb{E}_Q\left[d\left(\mathbb{P}_{Y|Q},Q\right)\right] + \mathbb{E}_Q\left[H(\mathbb{P}_{Y|Q})\right] \;. \tag{14}$$

It remains to be shown that $\mathbb{E}_Q\left[H(\mathbb{P}_{Y|Q})\right] = I(Y;X|Q) + \mathbb{E}_X\left[H(\mathbb{P}_{Y|X})\right]$.

Note that $\mathbb{P}_{Y|X} = \mathbb{P}_{Y|Q,X}$ under the very natural assumption that the model and the target are conditionally independent given the feature (i.e. there is no side information). Again using the law of total expectation, we have

$$\mathbb{E}_Q\left[H(\mathbb{P}_{Y|Q})\right] = \mathbb{E}_Q\left[\mathbb{E}_{X|Q}\left[H(\mathbb{P}_{Y|Q})\mid Q\right]\right] \tag{15}$$

$$= \mathbb{E}_Q\left[\mathbb{E}_{X|Q}\left[H(\mathbb{P}_{Y|Q}) - H(\mathbb{P}_{Y|X}) + H(\mathbb{P}_{Y|X})\mid Q\right]\right] \tag{16}$$

$$= \mathbb{E}_Q\left[\mathbb{E}_{X|Q}\left[H(\mathbb{P}_{Y|Q}) - H(\mathbb{P}_{Y|X})\mid Q\right]\right] + \mathbb{E}_Q\left[\mathbb{E}_{X|Q}\left[H(\mathbb{P}_{Y|X})\mid Q\right]\right] \tag{17}$$

$$= \mathbb{E}_Q\left[H(\mathbb{P}_{Y|Q}) - \mathbb{E}_{X|Q}\left[H(\mathbb{P}_{Y|X})\mid Q\right]\right] + \mathbb{E}_Q\left[\mathbb{E}_{X|Q}\left[H(\mathbb{P}_{Y|X})\mid Q\right]\right] \tag{18}$$

$$= \underbrace{\mathbb{E}_Q\left[H(\mathbb{P}_{Y|Q}) - \mathbb{E}_{X|Q}\left[H(\mathbb{P}_{Y|Q,X})\mid Q\right]\right]}_{I(Y;X|Q)} + \mathbb{E}_X\left[H(\mathbb{P}_{Y|X})\right] \;, \tag{19}$$

which concludes the proof. $\qquad\square$

## A.2. Proof of Proposition 5.2

*Proof.* In general any model is calibrated if it is in the form of a conditional law $\mathbb{P}_{Y|\phi(X)}$, where $\phi$ is an arbitrary measurable function. Let $Z = \phi(X)$ and $W = \mathbb{P}_{Y|Z} = \psi(Z)$. Then we have

$$\mathbb{P}_{Y|W} = \mathbb{E}_{Z|W}\left[\mathbb{P}_{Y|Z}\mid W\right] = \mathbb{E}_{Z|W}\left[W\mid W\right] = W \;.$$

Choosing $\phi(X) = Q|X$ concludes the first part of the proof.

Under the very natural assumption that $Q \perp\!\!\!\perp Y$ given $X$, we have $\mathbb{P}_{Y|Q,X} = \mathbb{P}_{Y|\widetilde{Q},X} = \mathbb{P}_{Y|X}$. Therefore, we only have to show that

$$\mathbb{E}_Q\left[H(\mathbb{P}_{Y|Q})\right] = \mathbb{E}_{\widetilde{Q}}\left[H(\mathbb{P}_{Y|\widetilde{Q}})\right] \;, \tag{20}$$

in order to prove the right-hand side of Equation (8). From Definition 5.1 and the first part of the proof we have

$$\mathbb{P}_{Y|Q} = \widetilde{Q} = \mathbb{P}_{Y|\widetilde{Q}} \;. \tag{21}$$

Using Equation (21) and the total law of expectation, it is straightforward to verify Equation (20). $\qquad\square$

## A.3. Properties of Generalized Entropy and Mutual Information

We include Definition A.1, Proposition A.2 and A.3 only for completeness. They can be readily found e.g. in (Gneiting & Raftery, 2007; Dawid & Musio, 2014). The only novelty is Proposition A.4, which follows straightforwardly from Proposition A.3.

**Definition A.1.** A function $H : \mathcal{M}(\mathcal{Y}) \to \mathbb{R}$ is *concave*, if for all $\lambda \in [0,1]$ and any $q_1, q_2 \in \mathcal{M}(\mathcal{Y})$ we have

$$\lambda H(q_1) + (1-\lambda)H(q_2) \leq H(\lambda q_1 + (1-\lambda)q_2) \;, \tag{22}$$

where $\lambda q_1 + (1-\lambda)q_2$ is understood as a mixture distribution. We call $H$ *strictly concave* if (22) is satisfied with strict inequality for $\lambda \in (0,1)$.

**Proposition A.2.** *The generalized entropy (Equation (3)) is concave, and is strictly concave, when the underlying proper scoring rule is strictly proper.*

*Proof.* Let $q_1, q_2 \in \mathcal{M}(\mathcal{Y})$ and $\lambda \in [0, 1]$ be arbitrary. Define $q = \lambda q_1 + (1 - \lambda) q_2$. Observe that

$$\mathbb{E}_{M \sim q_1} [S(q_1, M)] \leq \mathbb{E}_{M \sim q_1} [S(q, M)] \tag{23}$$

$$\mathbb{E}_{M \sim q_2} [S(q_2, M)] \leq \mathbb{E}_{M \sim q_2} [S(q, M)] \ , \tag{24}$$

since $S$ is (strictly) proper. Now add the (23) and (24) inequalities together multiplied by weights $\lambda$ and $(1 - \lambda)$ respectively and observe that:

$$\lambda \mathbb{E}_{M \sim q_1} [S(q_1, M)] + (1 - \lambda) \mathbb{E}_{M \sim q_2} [S(q_2, M)] \leq \mathbb{E}_{M \sim q} [S(q, M)] \ ,$$

since by the linearity of the expectation operator we have $\lambda \mathbb{E}_{M \sim q_1} + (1 - \lambda) \mathbb{E}_{M \sim q_2} = \mathbb{E}_{M \sim q}$.

$\square$

Let us state an important property of the entropy function $H$, which enables us to measure the dependence of two random variables via a general scoring rule.

**Proposition A.3.** *Let $\xi$ and $\eta$ be jointly distributed random variables. The generalized entropy is monotone, that is*

$$H(\mathbb{P}_\xi) \geq \mathbb{E}_{\eta \sim \mathbb{P}_\eta} \left[ H(\mathbb{P}_{\xi|\eta}) \right] \ . \tag{25}$$

*When using a strictly proper scoring rule, there is equality in (25) iff $\xi$ and $\eta$ are independent.*

*Proof.* Observe that $\mathbb{P}_\xi = \mathbb{E}_{\eta \sim \mathbb{P}_\eta} \left[ \mathbb{P}_{\xi|\eta} \right]$ is a convex mixture. Since $H$ is (strictly) concave, by the Jensen inequality we have

$$H \left( \mathbb{E}_{\eta \sim \mathbb{P}_\eta} \left[ \mathbb{P}_{\xi|\eta} \right] \right) \geq \mathbb{E}_{\eta \sim \mathbb{P}_\eta} \left[ H(\mathbb{P}_{\xi|\eta}) \right] \ . \tag{26}$$

If there is an event with nonzero $\mathbb{P}_\eta$ probability, where $\mathbb{P}_{\xi|\eta}$ differs from $\mathbb{P}_\xi$, then by the strict concavity of $H$, we will get a strict inequality in (26). $\square$

Based on Proposition A.3, one define the generalized mutual information of random variables $\xi, \eta$ as

$$I(\xi; \eta) = H(\mathbb{P}_\xi) - \mathbb{E}_{\eta \sim \mathbb{P}_\eta} \left[ H(\mathbb{P}_{\xi|\eta}) \right] \ , \tag{27}$$

i.e., via the amount of expected entropy reduction of $\xi$, if we condition on $\eta$. If $S$ is strictly proper, then $I(\xi, \eta) = 0$ implies that $\xi$ is independent of $\eta$. (Dawid & Musio, 2014)

For convenience, we will introduce the generalized *conditional* mutual information induced by $S$ as

$$I(\xi; \eta | \zeta) = \mathbb{E}_{\zeta \sim \mathbb{P}_\zeta} \left[ H(\mathbb{P}_{\xi|\zeta}) - \mathbb{E}_{\eta \sim \mathbb{P}_{\eta|\zeta}} \left[ H(\mathbb{P}_{\xi|(\eta, \zeta)}) \right] \right] \ . \tag{28}$$

**Proposition A.4.** *Given a strictly proper scoring rule, the generalized conditional mutual information (Equation (28)) is nonnegative and characterizes conditional independence, i.e.*

$$I(\xi; \eta | \zeta) = 0 \iff \xi | \zeta \perp\!\!\!\perp \eta | \zeta \quad \text{with } \mathbb{P}_\zeta \text{ probability } 1 \ .$$

*Proof.* Conditioning on fixed events $\{\zeta = \zeta_0\}$, Proposition A.3 can be applied point-wise. Taking the expectation with respect to $\mathbb{E}_\zeta$ results in having the non-negativity and characterization of conditional independence $\mathbb{P}_\zeta$ almost everywhere.

$\square$

## B. Connection of $I(Y; X|Q)$ and the Grouping Loss and a Formal Presentation of the Calibration-Sharpness Paradigm

Kull & Flach (2015) defined the following score decomposition[4]

$$\mathbb{E}\left[S(Q, Y)\right] = \underbrace{\mathbb{E}\left[d\left(\mathbb{P}_{Y|Q}, Q\right)\right]}_{\text{calibration error}} + \underbrace{\mathbb{E}\left[d\left(\mathbb{P}_{Y|X}, \mathbb{P}_{Y|Q}\right)\right]}_{\text{grouping loss}} + \underbrace{\mathbb{E}\left[H(\mathbb{P}_{Y|X})\right]}_{\text{aleatoric uncertainty}} . \tag{29}$$

Although they only considered classification models, their result remains valid in the regression setup as well. Using the definition of score divergence (Eq. 2), conditional mutual information (Def. 3.1) and the natural modeling assumption that $\mathbb{P}_{Y|Q,X} = \mathbb{P}_{Y|X}$ it is straightforward to see that

$$\underbrace{\mathbb{E}\left[d\left(\mathbb{P}_{Y|X}, \mathbb{P}_{Y|Q}\right)\right]}_{\text{grouping loss}} = \underbrace{I(Y; X|Q)}_{\text{lack of sharpness}} . \tag{30}$$

This means that our formalism for decomposing the sharpness part of the expected error quantitatively matches the notion of Kull & Flach (2015). Since the notion of mutual information contributes to the interpretability of the phenomena, we argue that our decomposition remains valuable.

It is also interesting to point out that the paradigm of *maximizing sharpness subject to calibration* can be formally shown to be equivalent with standard expected score minimization. In the original work of Gneiting et al. (2007), this was only stated as a conjecture due to the different notion of calibration used.

**Proposition B.1.** *Suppose that the model hypothesis space contains the true data generating process. Then the standard optimization problem*

$$\min_{Q} \mathbb{E}\left[S(Q, Y)\right] \tag{31}$$

*has the same unique solution as the following constrained optimization*

$$\min_{Q} \mathbb{E}\left[H(Q)\right] \quad \text{subject to} \quad \mathbb{P}_{Y|Q} = Q . \tag{32}$$

*Proof.* Let $S$ be a strictly proper scoring rule. We know that the unique minimum of (31) is obtained at the true target distribution $Q^{\text{opt}} = \mathbb{P}_{Y|X}$. Consider a model $Q$ that is a feasible solution to (32). Then using Equation (7), the expected score has 0 calibration error and the lack of sharpness and aleatoric uncertainty can be expressed as the expected predictive entropy, i.e.,

$$\mathbb{E}\left[S(Q, Y)\right] = \underbrace{I(Y; X|Q)}_{\text{lack of sharpness}} + \underbrace{\mathbb{E}\left[H(\mathbb{P}_{Y|X})\right]}_{\text{aleatoric uncertainty}} \tag{33}$$

$$= \mathbb{E}\left[H(\mathbb{P}_{Y|Q}) - H(\mathbb{P}_{Y|X})\right] + \mathbb{E}\left[H(\mathbb{P}_{Y|X})\right] \tag{34}$$

$$= \mathbb{E}\left[H(\mathbb{P}_{Y|Q})\right] \tag{35}$$

$$= \mathbb{E}\left[H(Q)\right] . \tag{36}$$

We know that $Q^{\text{opt}}$ is the unique minimizer of the expected score, and since it is Auto-calibrated[5], it also uniquely minimizes $\mathbb{E}\left[H(Q)\right]$ subject to calibration. $\qquad\square$

In light of Proposition B.1, we can readily see that under the calibration constraint, lack of sharpness becomes more concrete than the mutual information-based notion: it equals to the lack of *predictive* sharpness in terms of expected entropy, i.e.,

$$I(Y; X|Q) = \mathbb{E}\left[H(Q)\right] - \mathbb{E}\left[H(\mathbb{P}_{Y|X})\right] . \tag{37}$$

---

[4]We adapted their notion to match the presentation of our paper.
[5]E.g., see the proof of Proposition 5.2.

# C. Distributional Pre-image Problem

Conditional kernel mean embeddings only estimate $\hat{\mu}_{Y|X} \approx \mathbb{E}[\psi(Y)|X]$, which is a representation of $\hat{\mathbb{P}}_{Y|X}$ not necessarily easy to work with. To be able to quantify the model error $\mathbb{E}\left[S(\hat{\mathbb{P}}_{Y|X}, Y)\right]$ or have predictive quantiles $\hat{\mathbb{P}}(Y \leq t)$, one often needs a more exact form of $\hat{\mathbb{P}}_{Y|X}$. This problem is called the distributional pre-image problem, since we are interested in finding the distribution $q \in \mathcal{M}(\mathcal{Y})$ whose kernel mean embedding is $\hat{\mu}_{Y|X=x}$. There are several different approaches to solve this problem (Muandet et al., 2017; Chen et al., 2012; Schuster et al., 2020). In order to minimize computational complexity and approximation bias, we have chosen the following approximate pre-image approach.

The approximate distributional pre-image solution starts with choosing a family of parameterized distributions $\{q_\theta \mid \theta \in \Theta\} \subset \mathcal{M}(\mathcal{Y})$ and define the approximate pre-image $\widetilde{\mathbb{P}}_{Y|X=x}$ as

$$\widetilde{\mathbb{P}}_{Y|X=x} = \arg\min_{\theta \in \Theta} ||\hat{\mu}_{Y|X=x} - \mathbb{E}_{M \sim q_\theta}[\psi(M)]||_{\mathscr{H}_l}^2 . \tag{38}$$

If the parameterized family is too rich, then the optimization (38) can be challenging to solve, and if it is too restrictive then one introduces a significant approximation error to the predictions. We made the following practical choice: Let $n = |\mathcal{D}_{\text{cal}}|$, let $\Theta = \Delta_n$, i.e. the $n$ dimensional probability simplex, and let $q_\theta = \sum_{i=1}^n \theta_i \delta_{y_i}$ be the empirical distribution supported on the observations in the calibration set with weight $\theta_i$ for the point-mass $\delta_{y_i}$. This is a reasonable choice since extending the support of $\hat{\mathbb{P}}_{Y|X}$ beyond $\{y_i \mid i \in \mathcal{D}_{\text{cal}}\}$ requires prior knowledge (or assumptions) about $\mathbb{P}_{Y|X}$.

With this choice (38) is easy to show[6] to be equivalent with

$$\theta^* = \arg\min_{\theta \in \Delta_n} (\theta - \beta)^T L (\theta - \beta) , \tag{39}$$

where $\beta$ are the weights in the conditional kernel mean embedding estimate $\hat{\mu}_{Y|X=x} = \sum_{i=1}^n \beta_i \psi(y_i)$, $L$ is the kernel matrix $[l(y_i, y_j)]_{i,j=1}^n$ and $\widetilde{\mathbb{P}}_{Y|X=x}$ results to be $\sum_{i=1}^n \theta_i^* \delta_{y_i}$.

Although (39) is a convex problem and therefore can be solved efficiently for each observation $X = x$, solving simultaneously for all observations in $\mathcal{D}_{\text{test}}$ is computationally challenging. Consequently, we decided to further approximate $\theta^*$ with

$$\widetilde{\theta}^* = \arg\min_{\theta \in \Delta_n} (\theta - \beta)^T I_n (\theta - \beta), \tag{40}$$

which is essentially the Euclidean projection of $\beta$ to $\Delta_n$, for which there is an $\mathcal{O}(n)$ time algorithm (Duchi et al., 2008). Using $\widetilde{\theta}^*$ instead of $\theta^*$ should be considered an implementation choice, which is not unprecedented; for example, the authors of (Cevid et al., 2022) also used clipped and renormalized CKME weights for inference in their benchmark section.

# D. Empirical Comparison Against Conformal Prediction

In this section, we compare conformal prediction (CP) with our proposed recalibration framework. We use the split conformal prediction framework with $\psi(x) = |x - \frac{1}{2}|$ of Chernozhukov et al. (2021).

### D.1. Evaluation

We compare the CP intervals $[a, b] \subset \mathbb{R}$ at a fixed coverage level ($\alpha = 0.05$) against the interval

$$[\tilde{a}, \tilde{b}] := [q_{\alpha/2}, q_{1-\alpha/2}] \subset \mathbb{R}$$

derived from the predictive quantiles $q$ of the recalibrated model. We plot the marginal coverage level on the test set, and the relative average width of the predicted intervals:

$$\frac{\tilde{b} - \tilde{a}}{b - a}$$

In order to highlight that standard CP procedures, such as (Chernozhukov et al., 2021) only target marginal alignment of model errors, and therefore the *"error cancellation"* (i.e., over and underconfident predictions cancelling out on average)

---

[6]Using the reproducing property of $l$

effect can occur, we plot the distance correlation $\mathrm{dCor}$ (Székely et al., 2007) of the predicted interval (as a point in $\mathbb{R}^2$) and the pit transform of the prediction $Z = F(Y)$.

It is easy to see that $Z$ should ideally be independent of the prediction and therefore from the interval $[a, b]$. This independence condition means that the errors are evenly distributed w.r.t. the predictions and there are no systematically under / overconfident predictions.

Since distance correlation is a normalized dependence measure, characterizing independence (i.e., $\mathrm{dCor} = 0 \iff$ the inputs are independent, and $0 \leq \mathrm{dCor} \leq 1$) it is a suitable metric to assess the amount of dependence between predictive intervals and the PIT transform. *Smaller values of* $\mathrm{dCor}$ *indicates better calibration.*

### D.2. Conclusions

We can see that the proposed method's marginal coverage and interval lengths are comparable to those of standard CP, while the dependence between predicted intervals and the PIT transform of the observations tends to be smaller (as the dataset size increases), i.e., the remaining modeling error is more evenly distributed. See Figure 4 for our results on comparing predictive interval coverage, average interval length, and dependence of the PIT transformed observation on the interval. The source code for this experiment is also available at `https://github.com/adamgnuj/recalibration_icml2026`.

## E. An Ablation Study on the Efficiency of the EDK

We performed an ablation study on comparing the runtime efficiency of the proposed Energy Distance Kernel with a naive implementation of the *Gaussian type* kernel (cf. Equation (9)).

As can be seen from the algorithmic complexity of the two approaches, if we have $n$ predictions to compare and each prediction is an $m$-sample empirical distribution, then we reduce the complexity of building the kernel matrix from $\mathcal{O}(n^2 m^2)$ to $\mathcal{O}(nm \log(m) + n^2 m)$. This efficiency gain is possible because the evaluation of the Energy Distance Kernel requires only linear time in $m$ once the samples are sorted.

Please find our detailed results on Figure 5.

## F. A Direct Comparison Against Standard Kernelized Distribution Regression

We perform a direct comparison of our proposed recalibration algorithm and the baseline model of only using kernelized predictions (without recalibration), trained on the union of training and calibration data.

This motivates the two stage learning approach, as we can see that even kernel methods can have significant calibration errors. (See middle figure on Figure 6.) Additionally, because kernel methods sometimes struggle to capture the mapping $X \mapsto \mathbb{P}_{Y|X}$, it is reasonable to use more sophisticated machine learning algorithms followed by a separate recalibration phase. (See, e.g., datasets 'bike' and 'naval-propulsion-plant'.)

### F.1. Evaluation

We report the relative CRPS error of the models, normalized as a ratio to what the kernelized prediction achieves. Additionally, we report the ratio of splits where the SKCE test rejected the hypothesis of auto calibration. The PIT calibration hypothesis test's results can also be found in a similar presentation.

See results on Figure 6.

### F.2. Implementation Details

We used a Gaussian kernel on the input space, and optimized the input kernel bandwidth and regularization parameter via 5-fold cross validation on the union of training and calibration data. The initial guess for the input kernel bandwidth was the median heuristic, and then we searched a logarithmically spaced grid around it. The output kernel was the Laplacian kernel where the bandwidth was set using the median heuristic. The source code for this experiment can also be found at `https://github.com/adamgnuj/recalibration_icml2026`.

## G. Wall Clock Time Comparison Against PIT Recalibration

Please find a wall-clock time comparison of our CKME based recalibration method vs. PIT recalibration (Kuleshov et al., 2018) on Figure 7.

## H. Benchmark

### H.1. Synthetic Data Set

We sampled $n = 20\,000$ i.i.d. feature samples from $U[-1, 1]$ and then generated the corresponding target variable from the 2 component mixture model

$$\tfrac{1}{2}\mathcal{N}\left(x + \sin(3x/20), \sigma^2\right) + \tfrac{1}{2}\mathcal{N}\left(x - \sin(3x/20), \sigma^2\right) \;,$$

where the variance of both components were $\sigma^2 = \frac{1}{10^2}$.

### H.2. Detailed Benchmark Results

Find detailed benchmark results of the relative CRPS scores in Table 1. Figure 3 contains aggregated results of acceptance rate (at $\alpha = 5\%$) for the hypothesis of PIT-calibration. See Figure 8, 9, 10 and 11 for detailed results of calibration hypothesis tests (where the relative CRPS results are also plotted, for easier comparison.)

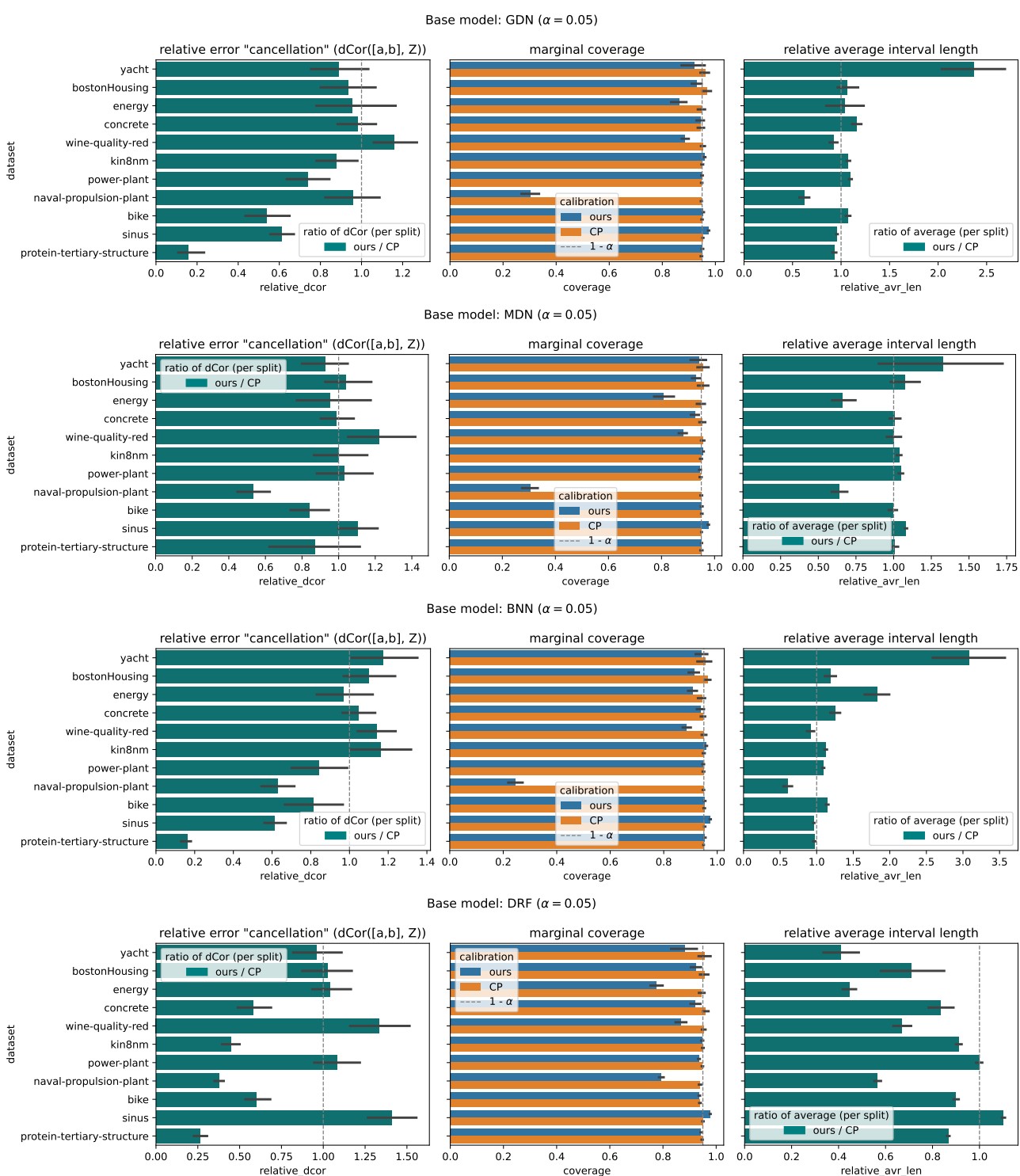

*Figure 4.* Benchmark results for comparing our recalibration approach to conformal prediction. See Appendix D.

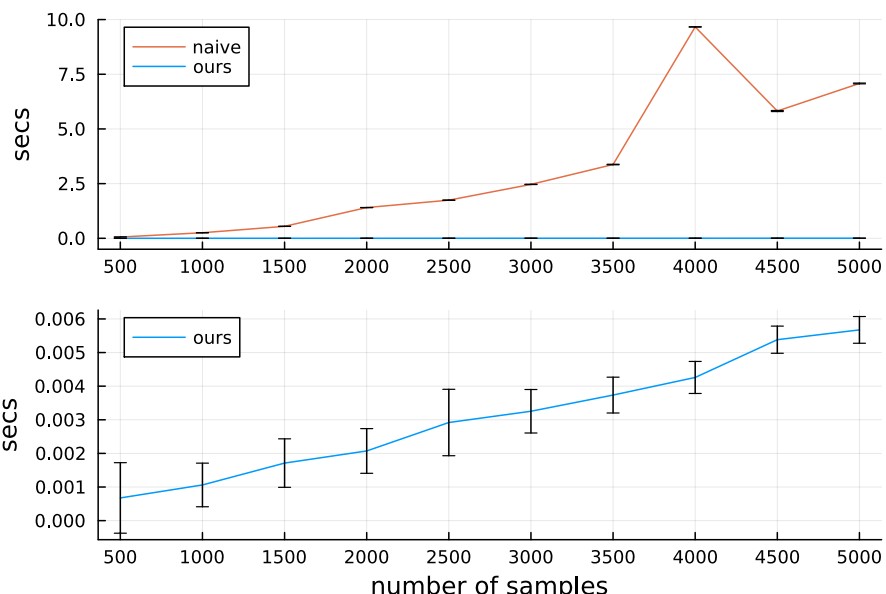

*Figure 5.* While the number of observations remains constant (at $n = 300$), we systematically vary the sample size used to represent the predicted distributions. Each configuration was executed for a minimum of $25$ trials. The cause of the performance spike at a sample size of $4000$ remains unclear; it may be an artifact of the GPU parallelization setup. Since kernel matrix construction relies mostly on calculating pairwise distances, our timing benchmarks focus solely on the distance matrix computation. All experiments were conducted on an NVIDIA A100 GPU. (See Appendix E.)

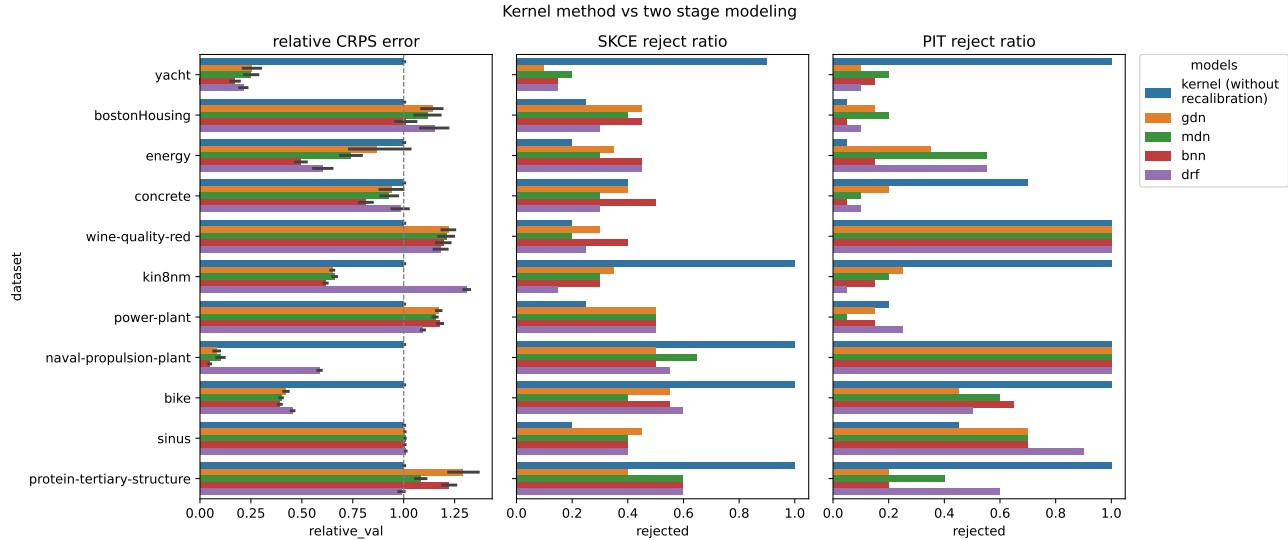

*Figure 6.* Results on a direct comparison of our two stage recalibration framework against standard kernelized distribution regression. (See Appendix F.)

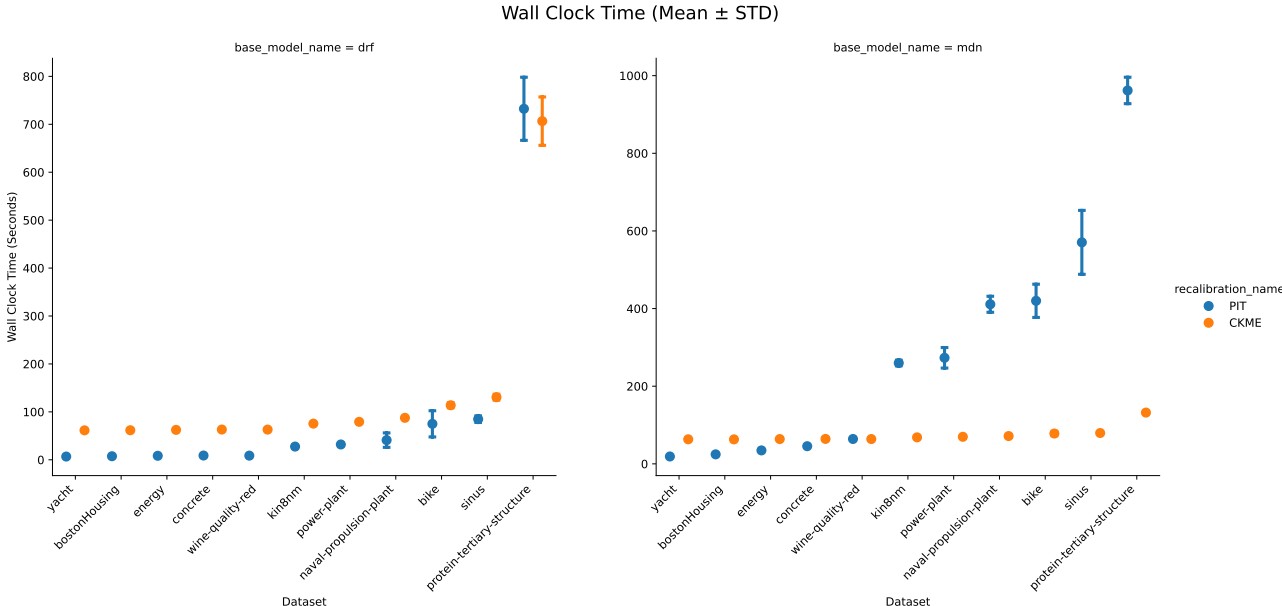

*Figure 7.* The relative latency of the PIT method stems from our implementation requirement to explicitly represent recalibrated predictive distributions for testing. For base models producing a continuous PDF (e.g., MDN), we utilized inverse CDF sampling with $n = 1000$. In cases where the base model's output was concentrated on discrete observations (e.g., DRF), we achieved PIT calibration via explicit weight transformation. All linear algebra operations for CKME recalibration were accelerated on an NVIDIA A100 GPU. (See Appendix G.)

*Table 1.* CRPS scores relative to the base model trained only on the train set (None(T)). Best relative score per row is marked with **bold** values. Standard deviations per split are displayed for each entry. The numbers after the dataset name indicate the size of the test set $|\mathcal{D}_{\text{test}}|$.

| BASE MODEL | RE-CALIBRATION DATA SET | NONE(T) | NONE(T+C) | CKME | PIT | GPBETA |
|---|---|---|---|---|---|---|
| GDN | YACHT(31) | 1.000± 0.000 | **0.690**± 0.298 | 1.276± 0.537 | 0.997± 0.077 | 0.905± 0.198 |
| | HOUSING(51) | 1.000± 0.000 | **0.955**± 0.134 | 1.150± 0.170 | 1.000± 0.033 | 0.995± 0.096 |
| | ENERGY(77) | 1.000± 0.000 | 0.962± 0.679 | **0.767**± 0.224 | 0.981± 0.048 | 0.984± 0.048 |
| | CONCRETE(103) | 1.000± 0.000 | **0.968**± 0.139 | 1.045± 0.047 | 0.999± 0.014 | 1.000± 0.021 |
| | WINE(160) | 1.000± 0.000 | 0.996± 0.018 | **0.901**± 0.026 | 0.999± 0.009 | 1.000± 0.011 |
| | KIN8NM(819) | 1.000± 0.000 | **0.971**± 0.026 | 0.994± 0.012 | 0.996± 0.005 | 1.001± 0.006 |
| | POWER(957) | 1.000± 0.000 | **0.982**± 0.024 | 1.004± 0.007 | 0.999± 0.003 | 1.001± 0.004 |
| | NAVAL(1193) | 1.000± 0.000 | 1.104± 0.449 | **0.417**± 0.105 | 0.882± 0.086 | 0.896± 0.090 |
| | BIKE(1738) | 1.000± 0.000 | **0.950**± 0.076 | 0.977± 0.023 | 1.000± 0.007 | 1.003± 0.006 |
| | SINUS(2000) | 1.000± 0.000 | 1.000± 0.002 | **0.900**± 0.003 | 0.905± 0.002 | 1.033± 0.045 |
| | PROTEIN(4573) | 1.000± 0.000 | 0.975± 0.088 | **0.937**± 0.008 | 0.995± 0.005 | 0.991± 0.008 |
| MDN | YACHT(31) | 1.000± 0.000 | 0.960± 0.365 | 1.326± 0.591 | 1.058± 0.130 | **0.870**± 0.131 |
| | HOUSING(51) | 1.000± 0.000 | **0.944**± 0.114 | 1.175± 0.149 | 1.007± 0.030 | 1.027± 0.064 |
| | ENERGY(77) | 1.000± 0.000 | 0.794± 0.308 | **0.594**± 0.178 | 1.019± 0.046 | 1.153± 0.071 |
| | CONCRETE(103) | 1.000± 0.000 | **0.934**± 0.088 | 1.032± 0.038 | 1.003± 0.018 | 1.006± 0.021 |
| | WINE(160) | **1.000**± 0.000 | 1.005± 0.022 | 1.015± 0.023 | 1.020± 0.018 | 1.128± 0.027 |
| | KIN8NM(819) | 1.000± 0.000 | **0.962**± 0.041 | 1.001± 0.013 | 0.997± 0.008 | 0.999± 0.013 |
| | POWER(957) | 1.000± 0.000 | **0.986**± 0.008 | 1.001± 0.008 | 1.000± 0.002 | 1.017± 0.006 |
| | NAVAL(1193) | 1.000± 0.000 | 1.002± 0.344 | **0.455**± 0.102 | 0.907± 0.053 | 0.933± 0.054 |
| | BIKE(1738) | 1.000± 0.000 | **0.947**± 0.038 | 0.991± 0.016 | 1.001± 0.004 | 1.029± 0.009 |
| | SINUS(2000) | 1.000± 0.000 | **0.999**± 0.003 | 1.006± 0.004 | 1.000± 0.002 | 1.150± 0.045 |
| | PROTEIN(4573) | 1.000± 0.000 | **0.957**± 0.042 | 0.985± 0.005 | 1.000± 0.002 | 1.079± 0.013 |
| BNN | YACHT(31) | 1.000± 0.000 | 0.888± 0.171 | 1.614± 0.477 | 0.931± 0.117 | **0.848**± 0.141 |
| | HOUSING(51) | 1.000± 0.000 | **0.994**± 0.047 | 1.188± 0.143 | 1.001± 0.022 | 1.018± 0.039 |
| | ENERGY(77) | 1.000± 0.000 | **0.946**± 0.175 | 1.077± 0.187 | 0.977± 0.038 | 0.973± 0.059 |
| | CONCRETE(103) | 1.000± 0.000 | **0.960**± 0.074 | 1.065± 0.039 | 1.001± 0.015 | 1.011± 0.022 |
| | WINE(160) | 1.000± 0.000 | 0.989± 0.027 | **0.899**± 0.029 | 1.001± 0.014 | 1.005± 0.016 |
| | KIN8NM(819) | 1.000± 0.000 | **0.968**± 0.017 | 1.015± 0.012 | 0.999± 0.004 | 1.006± 0.009 |
| | POWER(957) | 1.000± 0.000 | **0.981**± 0.019 | 1.004± 0.007 | 0.999± 0.002 | 1.004± 0.005 |
| | NAVAL(1193) | 1.000± 0.000 | 0.975± 0.217 | **0.376**± 0.067 | 0.870± 0.046 | 0.889± 0.046 |
| | BIKE(1738) | 1.000± 0.000 | **0.977**± 0.040 | 1.003± 0.010 | 1.000± 0.002 | 1.005± 0.003 |
| | SINUS(2000) | 1.000± 0.000 | 0.999± 0.002 | **0.900**± 0.002 | 0.907± 0.004 | 1.030± 0.034 |
| | PROTEIN(4573) | 1.000± 0.000 | 1.002± 0.051 | **0.947**± 0.006 | 0.998± 0.002 | 0.993± 0.008 |
| DRF | YACHT(31) | 1.000± 0.000 | 0.814± 0.046 | **0.405**± 0.086 | 0.850± 0.160 | 0.885± 0.147 |
| | HOUSING(51) | 1.000± 0.000 | **0.949**± 0.022 | 0.984± 0.088 | 0.983± 0.034 | 0.964± 0.057 |
| | ENERGY(77) | 1.000± 0.000 | 0.809± 0.025 | **0.473**± 0.078 | 0.950± 0.040 | 0.985± 0.057 |
| | CONCRETE(103) | 1.000± 0.000 | 0.942± 0.013 | **0.871**± 0.045 | 0.943± 0.022 | 0.899± 0.049 |
| | WINE(160) | 1.000± 0.000 | **0.981**± 0.006 | 0.988± 0.025 | 1.404± 0.122 | 1.126± 0.024 |
| | KIN8NM(819) | 1.000± 0.000 | 0.979± 0.004 | **0.882**± 0.015 | 0.989± 0.004 | 1.003± 0.016 |
| | POWER(957) | 1.000± 0.000 | 0.975± 0.002 | **0.969**± 0.009 | 0.992± 0.002 | 1.017± 0.005 |
| | NAVAL(1193) | 1.000± 0.000 | 0.893± 0.006 | **0.560**± 0.016 | 0.792± 0.018 | 0.882± 0.033 |
| | BIKE(1738) | 1.000± 0.000 | 0.955± 0.006 | **0.918**± 0.010 | 0.966± 0.006 | 1.009± 0.007 |
| | SINUS(2000) | 1.000± 0.000 | 0.999± 0.003 | **0.997**± 0.004 | 0.999± 0.002 | 1.131± 0.025 |
| | PROTEIN(4573) | 1.000± 0.000 | 0.973± 0.001 | **0.923**± 0.007 | 0.984± 0.000 | 1.092± 0.012 |

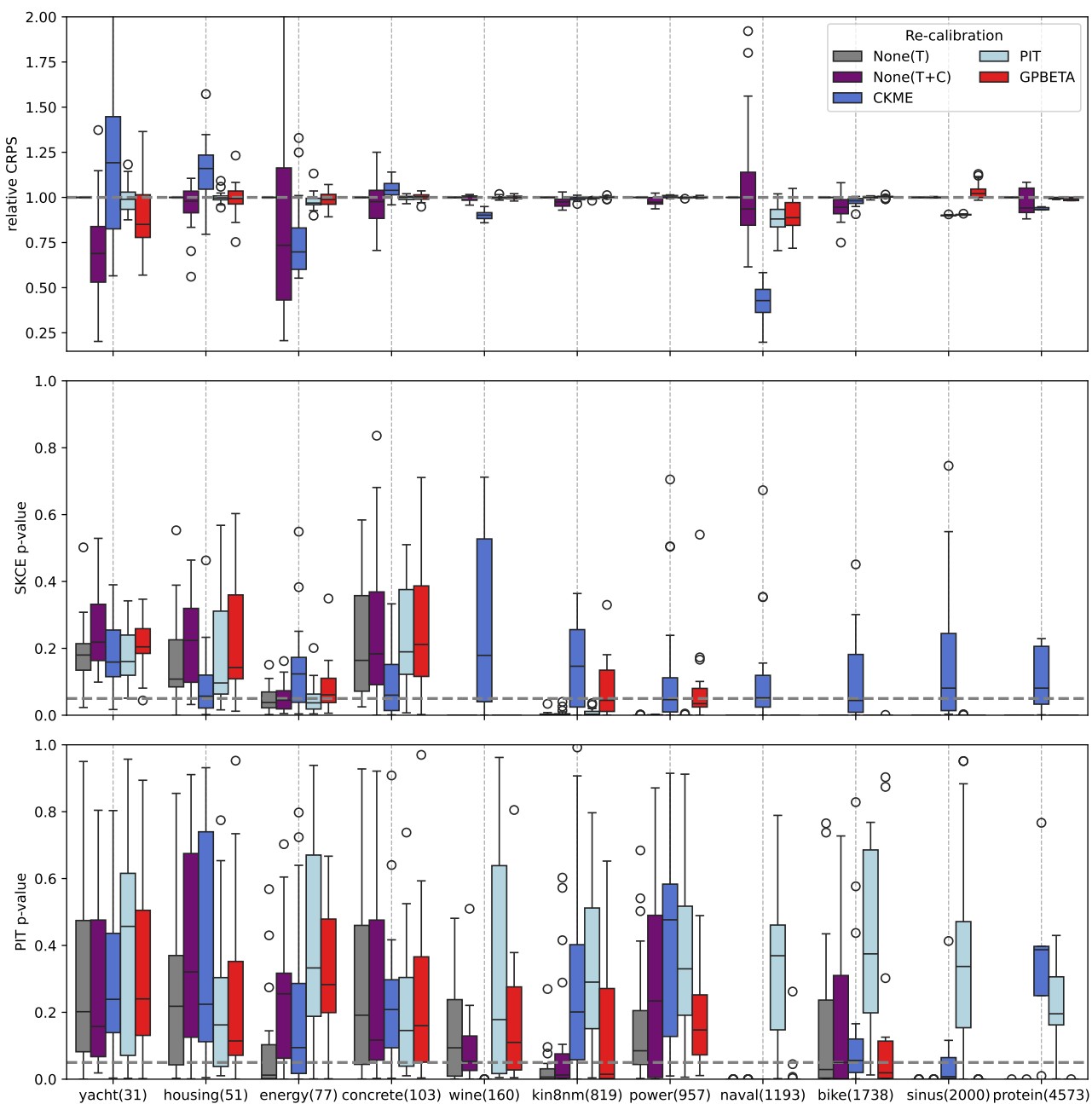

*Figure 8.* Detailed benchmark results for base model GDN.

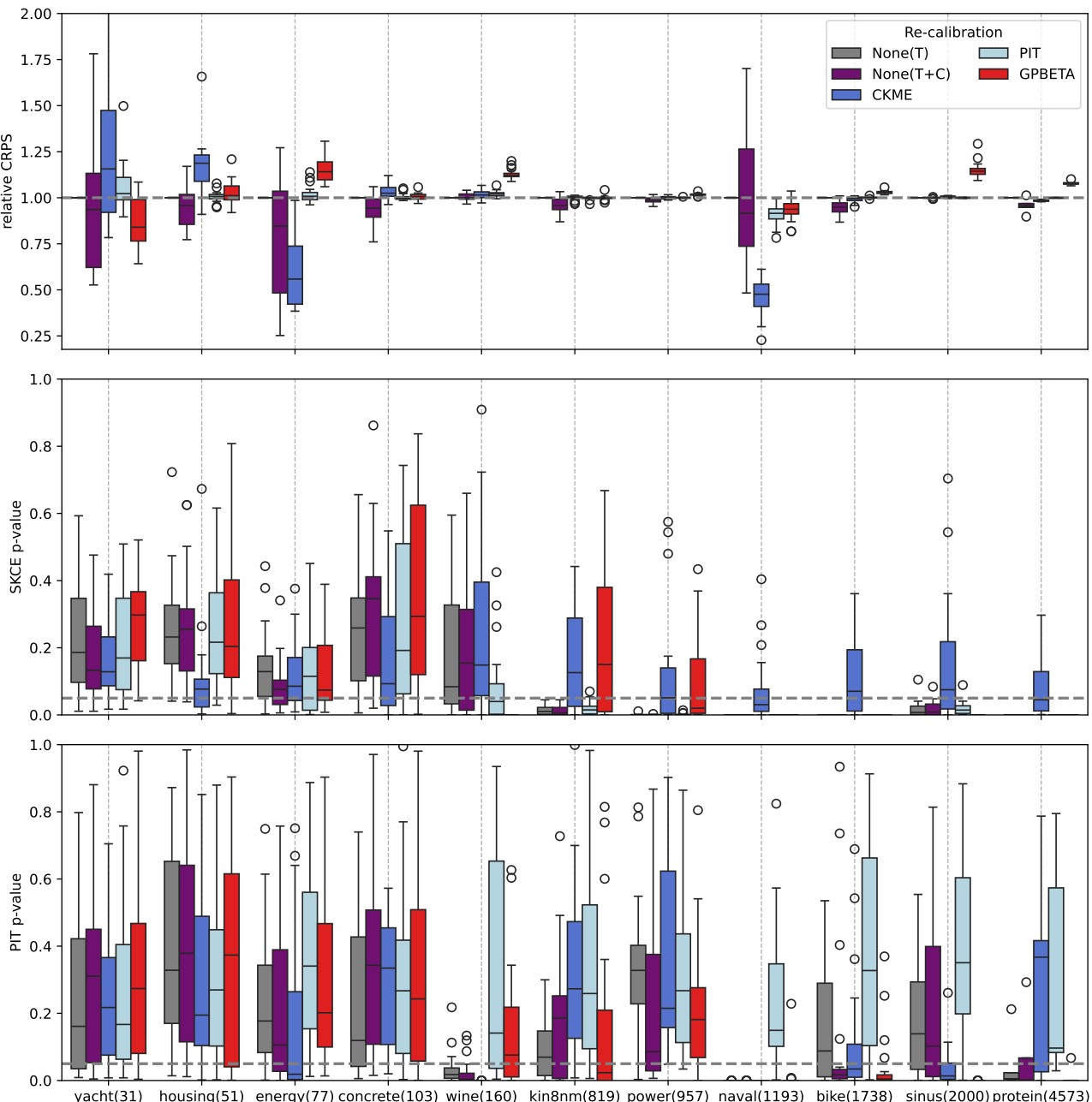

*Figure 9.* Detailed benchmark results for base model MDN.

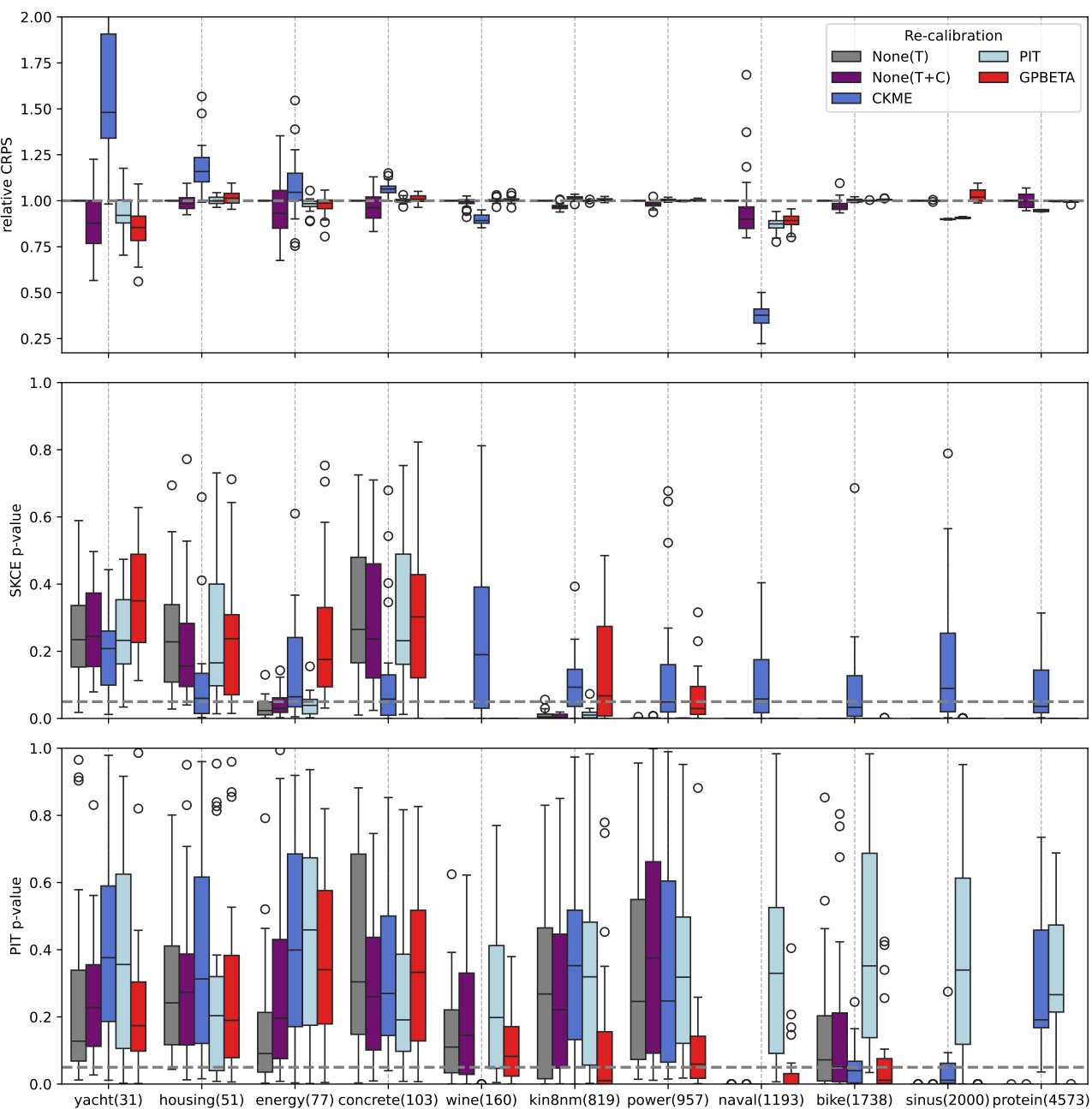

*Figure 10.* Detailed benchmark results for base model BNN.

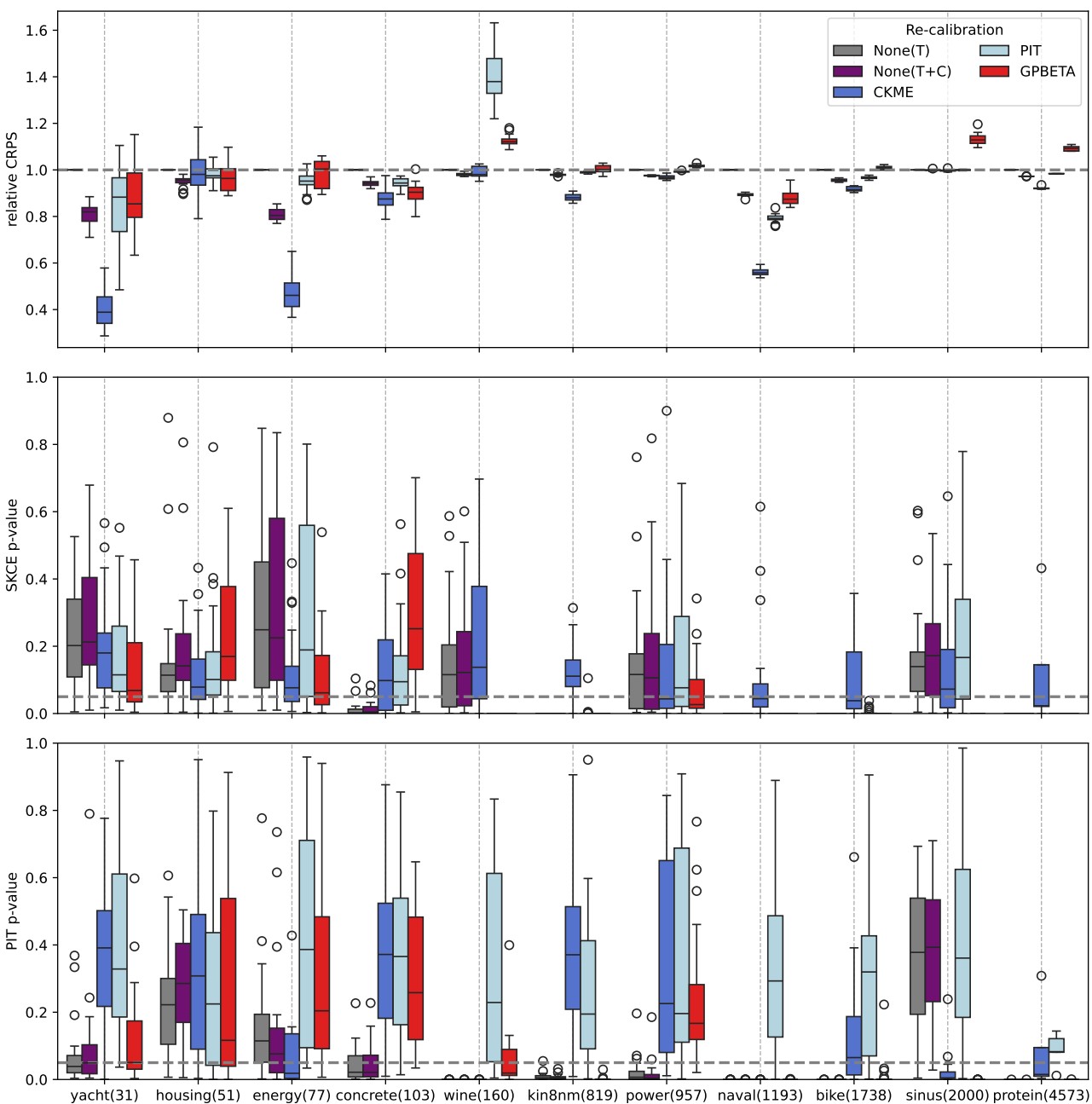

*Figure 11.* Detailed benchmark results for base model DRF.

