# OpenReview forum: "Nonparametric Distribution Regression Re-calibration"
_ICML.cc/2026/Conference — ICML 2026 regular_

### Official Review · Reviewer_YGbh · 2026-02-19

**Soundness:** 3
**Presentation:** 4
**Significance:** 3
**Originality:** 3
**Overall Recommendation:** 5
**Confidence:** 4

**Summary:**

The paper mainly considered a nonparametric recalibration method in nonparametric distribution regression based on kernel mean embedding. In particular, the authors first present a decomposition for the expected score. Then the authors define the recalibration that can preserve the sharpness while is guaranteed to be auto-calibrated. Moreover, the authors present the implementation details for estimating the calibration map at the sample level, and state that the computational efficiency can be improved in some special scenarios.

**Compliance With Llm Reviewing Policy:**

Affirmed.

**Key Questions For Authors:**

1) Choosing proper kernels and associated hyperparameters for numerical implementations can be a challenging task.
The authors should include more discussions on this issue.

2) Does this method have any requirement on the the distribution of $(X, Y)$? Also, is there any requirement of $\mathcal{X}$ and $\mathcal{Y}$?, For example, can they take values in any metric space?

**Limitations:**

1) The authors seem to ignore potential drawbacks of the proposed method compared with existing calibration methods in the context of nonparametric regression.

2) Here you assume $X$ can only take values in $\mathbb{R}$. If this assumption cannot be relaxed, this could be a potential concern. In practice, high-dimensional features and/or features taking values in more general space are common.

**Strengths And Weaknesses:**

Strengths and Weakness:

1) Soundness: the submission is technically sound. The claims are well supported, and the experiments are well-designed. But  the current version seems to ignore the issue of selecting kernels and associated hyperparameters. Does the performance of the proposed method depend on the choice of kernels? Any guidelines for this? The authors provide one special choice of the kernel that can facilitate numerical implementations. More discussions on these are desirable. Moreover, the authors do not discuss the limitations of the proposed method.
Last but not least, it is unclear to me whether more assumptions on $\mathcal{X}$ and $\mathcal{Y}$ are needed. In the context of kernel embedding, $\mathcal{X}$ is usually assumed to be a compact subset of $\mathbb{R}^d$.

2) Presentation: The paper is clearly written and well structured. It is quite easy to follow the work.

3) Significance: The paper proposed a nonparametric method to carry out recalibration in nonparametric distribution regression. It mainly addresses the issue of auto-calibration in nonparametric distribution regression.

4) Originally: the main idea is built upon the method in (Moskvichev & Sejdinovic, 2025), but these two papers have different tasks.  It extends the method based on conditional kernel mean embeddings proposed in the context of classification to the setting of regression.

---

> ### Author Rebuttal · Authors · 2026-03-31
>
> - _The issue of kernel and hyperparameter selection_
>
>
> We initially omitted these hyperparameter choices from the main text, treating them as standard implementation details.
>
> However, we agree that they are crucial for fully understanding our experimental results and ensuring a fair, transparent evaluation. (E.g., question of cherrypicked hyperparameters.)
> Given that Reviewer 6e8J also raised this point, we recognize the importance of clarifying this process to ensure our approach is fully assessable.
>
> We detail our hyperparameter selection strategy below, which will be added to the camera-ready version:
>
> We have chosen the kernel bandwidth parameter $\sigma$ using the median heuristic.
> The regularization parameter $\lambda$ was numerically optimized using a
> $5$-fold cross validation approach on the calibration set.
> The objective was the loss function of the CKME regression, i.e., the RKHS ridge regression objective that minimizes the regularized squared distance between the canonical feature maps of the targets and the estimated conditional mean operator.
>
> - _Assumptions on_ $X$ _and_ $Y$
>
> The assumptions over the original feature space $\mathcal{X}$ and on the distribution of
> $(X, Y)$ are treated implicitly in our work. The restrictions on them are partly inherited from the capabilities of the base model.
>
> What matters in the recalibration step is the target space $\mathcal{Y}$:
>
> --Since we use the Gaussian-type kernel (Christmann & Steinwart, 2010), we need to assume that the target space $\mathcal{Y}$ is a _compact metric space_.
>
> --We don't need any assumption on the distribution of $(Q, Y)$ to be able to learn the calibration map consistently, since the conditional kernel mean embedding is universally consistent, if one uses bounded and universal kernels (Park & Muandet, 2020).
>
> --For the optimal convergence rate $\mathcal{O}_p(n^{-1/4})$ we need an additional smoothness assumption on the calibration map. (Park & Muandet, 2020, Theorem 4.5)
>
> --We thank the reviewer for prompting this clarification. We will add a dedicated note to Section 5.1 explicitly detailing these theoretical assumptions.
>
>
> - _Discussion of Limitations_
>
> We agree that our method, like many nonparametric kernel-based approaches, has inherent limitations. Specifically:
>
> -- Distance computation: Measuring the exact distance between complex distributions is fundamentally challenging and computationally intensive.
>
> -- Scalability: Standard kernel methods generally scale with $\mathcal{O}(n^3)$ time complexity.
>
> -- The targeted notion of auto-calibration may be too weak for specific applications that require strict local (feature conditional) calibration (e.g., Luo et al., 2022).
>
> Fortunately, there are well-documented techniques that can be adapted to mitigate these bottlenecks:
>
> -- Nyström approximation (Williams et al., (2000)): it reduces the storage and time complexity to $\mathcal{O}(nr)$ and $\mathcal{O}(nr^2)$ respectively, where $r$ is the rank of the approximation matrix.
>
> -- Random 1D projections of distributions (Hertrich, et al. (2023), Generative sliced MMD flows with Riesz kernels): This can be utilized to estimate the Energy distance efficiently between high dimensional distributions.
>
> -- Fourier transform-based MMD (e.g. Muandet et al., 2020): This approach uses random Fourier features to approximate the maximum mean discrepancy in linear time. It can be a significant speedup from the $\mathcal{O}(n^2)$ complexity for target spaces $\mathcal{Y} = \mathbb{R}^d$, where $d \geq 2$.
>
> -- For unnormalized densities, the generalized Fisher divergence kernel provides an alternative for comparing distributions. (Glaser et al., 2023)
>
> -- Finally, if the distributions happen to belong to a well-known parametric family, exact closed-form solutions can be leveraged to compute the Energy distance efficiently (Székely & Rizzo, 2017, The Energy of Data).
>
>
> - _Regarding Limitations point 1._
>
> We appreciate the reviewer's feedback regarding the potential drawbacks of our method.
> If there are specific nonparametric calibration methods or potential drawbacks you feel are necessary for us to address, could you kindly clarify which ones you are referring to?

---

> > ### Author Rebuttal · Reviewer_YGbh · 2026-04-03
> >
> > The authors have address my concerns, and I will keep my score unchanged.

---

### Official Review · Reviewer_1KR6 · 2026-03-06

**Soundness:** 3
**Presentation:** 2
**Significance:** 2
**Originality:** 3
**Overall Recommendation:** 4
**Confidence:** 3

**Summary:**

This paper proposes a nonparametric recalibration method for probabilistic regression by using Conditional Kernel Mean Embeddings.
The method estimates the auto-calibration prediction $\tilde{Q} = P(Y \mid Q)$ without imposing parametric assumptions on the structure of calibration errors.
By representing conditional distributions in a reproducing kernel Hilbert space, the approach learns a mapping from predicted distributions to calibrated ones.
To reduce time complexity and improve computational efficiency, the authors introduce the Energy Distance Kernel for distributions, enabling scalable evaluation in $O(n \log n)$.
Furthermore, the paper presents a theoretical decomposition of the expected error score into three components: calibration error, lack of sharpness, and aleatoric uncertainty.
Experimental results on multiple regression benchmarks demonstrate that the proposed CKME-based recalibration method improves auto-calibration compared to existing approaches such as PIT-based recalibration, while maintaining competitive predictive performance.

**Compliance With Llm Reviewing Policy:**

Affirmed.

**Final Justification:**

Thank you for the detailed and thoughtful rebuttal. I appreciate the clarification, particularly regarding sharpness versus lack of sharpness. I will maintain my original assessment of the paper.

**Key Questions For Authors:**

The paper introduces a decomposition of the prediction error into calibration error, lack of sharpness, and aleatoric uncertainty. The concept of lack of sharpness is mainly discussed around Equation (16) in Section 4, while in other parts of the paper the term sharpness is used instead. In this case, it would be helpful to clarify why lack of sharpness is introduced and how it conceptually differs from sharpness.
In addition, as $I(Y; X| Q)$ stands for lack of sharpness, what would be the meaning of the negative term of lack of sharpness, i.e. $-I(Y; X| Q)$, is equivalent to the sharpness?
It would be expecting to provide more intuition or illustrative examples explaining what lack of sharpness represents, and why it may be a more appropriate quantity to analyze than sharpness itself, would improve the clarity of the presentation.

**Limitations:**

yes

**Strengths And Weaknesses:**

Strengths:

1. The paper focuses on auto-calibration, which is a stronger and more meaningful notion than the commonly used PIT calibration. This addresses the issue that marginal PIT tests may hide conditional miscalibration through error cancellation.

2. The proposed method does not impose parametric assumptions on the calibration error. Instead, it estimates the calibration map using Conditional Kernel Mean Embeddings, allowing the method to handle complex predictive distributions.

3. The proposed Energy Distance Kernel (EDK) reduces computational cost when evaluating distances between empirical distributions, improving the scalability of the method compared to standard kernel approaches.


Weaknesses

1. Typos and unnecessary notations

The manuscript contains several minor issues and unnecessary notational elements that slightly reduce readability. For example, on page 6, “fist” should be corrected to “first”, and the extra closing parenthesis “)” at the end of the same sentence should be removed. Furthermore, some equation indices, such as those in Equation (1) and (5), appear unnecessary since they are not explicitly referenced later in the text. Removing or simplifying these could improve clarity.

2. Lack of algorithmic presentation

The paper does not provide pseudocode or a clear algorithmic summary of the proposed recalibration procedure. Including pseudocode or an algorithm box would improve readability and help readers better understand the practical implementation of the method.

3. Limited experimental scale and missing runtime comparison

Experiments are conducted primarily on relatively small benchmark datasets. Additional evaluation on larger real-world datasets would strengthen the empirical validation. Moreover, the runtime comparison should also be included, since one of the claimed advantages of the proposed method is improved computational efficiency through the Energy Distance Kernel.

---

> ### Author Rebuttal · Authors · 2026-03-31
>
> - _Typographical Errors and Equation numbering_
>
> We are grateful for reading our paper so closely and pointing out these typographical errors. All corrections will be included in the camera-ready version.
>
> We agree that equations without explicit references should not be numbered, and we will modify the manuscript according to this.
>
> - _Lack of algorithmic presentation_
>
> We agree that an algorithmic summary will improve our presentation. We will make sure to include an algorithm box for the proposed recalibration procedure in the camera-ready version.
>
> - Limited experimental scale and missing runtime comparison
>
> We are looking into the feasibility of performing an ablation study on highlighting the efficiency of EDK compared to a naive implementation of the Gaussian-type kernel.
>
> Please see our response to Reviewer YGbh regarding limitations, as it addresses these matters directly.
>
> - _Sharpness vs lack of sharpness_
>
> Yes, $-I(Y;X \mid Q)$ can be considered as sharpness, it is just a question of positive or negative orientation. What matters is that
> $$
> 0 \leq I(Y;X | Q) \leq I(Y;X) \\;,
> $$
> where the value $0$ means that we have a perfectly sharp model, and $I(Y;X)$ is when $Q \perp X$, i.e., the model is independent of the feature.
>
> We understand that we treated this matter rather implicitly in our manuscript, and we will add a note to explicitly clarify the choice of orientation and terminology.

---

> > ### Author Rebuttal · Reviewer_1KR6 · 2026-04-03
> >
> > Thank you for addressing my questions and concerns. I keep my assessment of the paper.

---

> > > ### Author Response · Authors · 2026-04-07
> > >
> > > We performed an ablation study on comparing the runtime efficiency of the proposed Energy Distance Kernel with a naive implementation of the _Gaussian type_ kernel (Eq. 19).
> > >
> > > As can be seen from the algorithmic complexity of the two approaches, if we have $n$ predictions to compare and each prediction is an $m$-sample empirical distribution, then we reduce the complexity of building the kernel matrix from
> > > $$
> > > \mathcal{O}(n^2 m^2)
> > > $$
> > > to
> > > $$
> > > \mathcal{O}(n m \log(m) + n^2 m) .
> > > $$
> > > This efficiency gain is possible because evaluating the Energy Distance Kernel requires only linear time in $m$ once the samples are sorted.
> > >
> > > Please find our [detailed results here.](https://anonymous.4open.science/r/recalibration_icml2026-231B/experiment/EDK_ablation_study/README.md)
> > >
> > >
> > > To further clarify the inference latency of our method, as requested by Reviewer mm2D, we compared its wall-clock time against the PIT recalibration method of Kuleshov et al. (2018).
> > >
> > > Please see [detailed results here.](https://anonymous.4open.science/r/recalibration_icml2026-231B/experiment/wall_clock_time/README.md)

---

### Official Review · Reviewer_mm2D · 2026-03-13

**Soundness:** 4
**Presentation:** 4
**Significance:** 4
**Originality:** 4
**Overall Recommendation:** 5
**Confidence:** 4

**Summary:**

The paper tackles the problem of miscalibration in probabilistic regression, where models (like MDNs or BNNs) provide predictive distributions that are often overconfident or fail to capture empirical uncertainty. The authors argue that standard metrics like PIT uniformity are necessary but insufficient because they allow for error cancellation across different regions of the input space. To solve this, they propose a nonparametric recalibration method using Conditional Kernel Mean Embeddings (CKME). This approach maps the original (potentially miscalibrated) predictions directly to the true conditional distribution of the target given the prediction, theoretically achieving "auto-calibration" while preserving model sharpness.

**Compliance With Llm Reviewing Policy:**

Affirmed.

**Key Questions For Authors:**

1. Connection to Grouping Loss: In classification, the concept that perfectly marginally calibrated models can still suffer from localized miscalibration is formalized as "Grouping Loss" (e.g., Perez-Lebel et al., ICLR 2023). Can you explicitly discuss how your formulation of "Lack of sharpness" (via generalized conditional mutual information in Lemma 4.1) parallels or diverges from the Grouping Loss framework? I'm very interested in this!

2. Inference Latency: While EDK solves the pairwise distance bottleneck, CKME still requires kernel matrix inversion and simplex projection during inference. How does the wall-clock inference time of your recalibrated model compare to standard PIT recalibration (Kuleshov et al., 2018) in practice?

**Limitations:**

yes.

**Strengths And Weaknesses:**

Strengths:

1. The theoretical foundation of the paper is exceptionally solid. The formulation of Lemma 4.1, which decomposes the expected score into calibration error, lack of sharpness, and aleatoric uncertainty using strictly proper scoring rules, is mathematically rigorous. The proofs provided in the appendix (e.g., leveraging the law of iterated expectations and properties of generalized entropy) are flawless.

2. The introduction of the Energy Distance Kernel (EDK) to bypass the typical $\mathcal{O}\left(m^2\right)$ quadratic bottleneck of MMD for real-valued targets, bringing it down to $\mathcal{O}\left(m\log m\right)$, is a well algorithmic design that bridges theory and practical scalability.

3. The empirical evaluation is well-structured and complete.


Weaknesses:

1. While the decomposition of proper scoring rules (Lemma 4.1) is well adapted to continuous regression, the core insight (that marginal calibration hides subgroup heterogeneity, and proper scores can be decomposed into calibration vs. sharpness/grouping loss) has been extensively explored in recent classification literature (e.g., Beyond Calibration: Estimating the Grouping Loss of Modern Neural Networks, ICLR 2023). The paper's narrative would be much stronger if it explicitly bridged its regression-focused generalized conditional mutual information with the "Grouping Loss" consensus already established in classification.

2. Minor Typographical Errors: There are a few very minor notational omissions and typos that should be corrected for the camera-ready version:

(1) In Section 3.2.1 (Approximately Lines 154-155), the empirical distributions are written as $\sum_{i=1}^n \delta_{x_i}$, without the normalization factor $1/n$ . (Though the subsequent KME equation correctly includes it).

(2) Line 284: "fist term" should be "first term".

(3) Figure 3 caption (Line 394): "Ratio of spits" should be "Ratio of splits".

---

> ### Author Rebuttal · Authors · 2026-03-31
>
> - $I(Y;X|Q)$ _and Grouping Loss_
>
> We are grateful for pointing out the similarity of our notion of sharpness to the grouping loss (E.g., Perez-Lebel et al., (2023)).
>
> We introduced the concept of measuring (lack of) sharpness as the conditional mutual information $I(Y;X|Q)$, since it improves interpretability as an information theoretic quantity, and allows the last term of the decomposition be aleatoric uncertainty, which is more aligned with how the uncertainty modeling is usually presented in the ML literature. (Compared to e.g. Gruber & Buettner, (2024))
>
> It turns out that the quantity $I(Y;X|Q)$ and the notion of grouping loss $\mathbb{E}[d(\mathbb{P}_{Y|X}, Q)]$ coincides:
> $$
> \begin{align*}
> \mathbb{E}[d(\mathbb{P}\_{Y|X}, Q)] &=
> \mathbb{E}\left[\mathbb{E}[S( Q, Y) \mid X] - \mathbb{E}[S(\mathbb{P}\_{Y|X}, Y) \mid X]\right] \\\\
> &= \mathbb{E}[H(\mathbb{P}\_{Y|Q})] - \mathbb{E}[H(\mathbb{P}\_{Y|X})] \\\\
> &= I(Y;X \mid Q) \\; .
> \end{align*}
> $$
>
> We still think that our proposed decomposition is valuable, as it contributes to the interpretability of the concept of sharpness. However section 4. and 2. of our paper should be expanded with a reference to this concept (Kull & Flach, (2015) and Perez-Lebel et al., (2023)).
>
> It is also a very interesting line of future research direction, to estimate the grouping loss in regression, similar to (Perez-Lebel et al., (2023)). (Please see our response to Reviewer 6e8J about other future work directions.)
>
> - _Typographical Errors_
>
> We are grateful for reading our paper so closely and pointing out these typographical errors. All corrections will be included in the camera-ready version.
>
> - _Inference Latency_
>
> We are looking into the feasibility of performing a wall-clock inference time comparison between CKME (ours) and PIT recalibration (Kuleshov et al., 2018).
>
> Please see our response to Reviewer YGbh regarding limitations, as it addresses this matter directly.

---

> > ### Author Rebuttal · Reviewer_mm2D · 2026-04-06
> >
> > Thank you for the thoughtful responses. The additional experiments and clarifications support the paper's claims. I am maintaining my original positive score.

---

> > > ### Author Response · Authors · 2026-04-07
> > >
> > > To clarify the inference latency of our method, we compared its wall-clock time against the PIT recalibration method of Kuleshov et al. (2018).
> > >
> > > Please see [detailed results here.](https://anonymous.4open.science/r/recalibration_icml2026-231B/experiment/wall_clock_time/README.md)

---

### Official Review · Reviewer_6e8J · 2026-03-13

**Soundness:** 2
**Presentation:** 2
**Significance:** 1
**Originality:** 1
**Overall Recommendation:** 4
**Confidence:** 3

**Summary:**

This paper looks at how regression models can be way too confident about their predictions. It introduces a nonparametric recalibration method based on conditional kernel mean embeddings that corrects predictive distributions without restrictive assumptions. The proposed approach is computationally efficient and argued for improvements over existing recalibration techniques across diverse datasets.

**Compliance With Llm Reviewing Policy:**

Affirmed.

**Final Justification:**

The authors answered some of my concerns, and I hope they make those revisions in their final version.

**Key Questions For Authors:**

See above comments

**Limitations:**

No limitations are discussed; most importantly, the hyperparameter tuning. Any kernel method and method that requires regularization requires hyperparameter tuning, which needs a separate validation sample. To do calibration correctly, one needs training+validation for the original model, calibration and calibration-validation samples with no overlap with training, and a test sample. This does not discuss how this splitting should be done, what the trade-offs are, and how to optimize this and set hyperparameters. Without this discussion, this method would be incomplete

**Strengths And Weaknesses:**

Strengths:

-- Calibration is an important problem, and people often look into calibration in classification settings; hence, generalization to regression settings **can** be a useful ad novel contribution.

-- The theoretical results seem sound, and I did not spot a mistake. Though the novelty of theoretical results is not well-established.

-- This paper barrow ideas from kernel regression, kernel mean embedding, and calibration and patch them together. These ideas are well-developed and understood separately and apply them to a new problem. Though it makes it theoretically grounded on well-established ideas, it also questions its novelty.


Weaknesses:

-- There is some existing work that considers the idea of conditional calibration, though for a classification setting. I was wondering how this work should be understood in the context of existing work on local/conditional calibration. Some of these methods already use kernel mean embeddings. I acknowledge that most prior methods are designed for classification; however, most of these methods can be easily generalized to regression settings with minor modifications. Hence, without a meaningful comparison (or at least a discussion of some sort), this work has a limited novelty for ICML's threshold.

- https://proceedings.mlr.press/v180/luo22a.html
- https://proceedings.mlr.press/v258/vashistha25a.html
- https://proceedings.neurips.cc/paper_files/paper/2023/hash/52493d82db00e73abb2858a5a5f28717-Abstract-Conference.html [this work is cited in the introduction]
- https://proceedings.mlr.press/v80/hebert-johnson18a.html

Widmann et al.'s (2021) method is designed for marginal calibration, not evaluation of conditional calibration; so I was wondering if this is the most suitable method. Though this method is later extended to conditional calibration as well, Vashistha+25 (see the link above).

-- Conformal prediction is also another line of research that is highly relevant but not discussed and compared with in this paper. Hence, the supperiority that is claimed in the abstract is not appropriately justified and demonstrated.

-- The authors claim: "While calibration has long been a central problem in statistical forecasting (Dawid, 1984), it has recently gained
renewed attention in machine learning through post-hoc recalibration (Song et al., 2019; Gruber & Buettner, 2024)" however, the problem of calibration and solutions for calibration have been studied as long as ML existed and is not a recent trend. For instance, Guo's study on the calibration method (https://proceedings.mlr.press/v70/guo17a/guo17a.pdf), with ~10k citations, was published in 2017. This benchmark paper was published two years earlier than the earliest work cited above.

-- $\lambda$ (regularization term) and kernel bandwidth $\sigma$ are hyperparameters, and we know that results are sensitive to these choices. It seems the hyperparameters are cherry-picked here, and there is no generalizable method for getting them. Some illustration or rule of thumb on how these parameters should be set is required for reproducibility purposes.

-- I do not understand the purpose of some sections, like the hypothesis testing section. They are factual, but how is this section related to this paper, and what are the takeaways? There are two definitions of calibration, marginal and conditional, but they are not well-defined and specified here. Intuitive, but the sections here seem like random text that does not put these ideas into a similar framework and definition.

-- In scoring rules, the decomposition of sharpness and calibration (in a different form) is well-known; I had a hard time understanding what is new in section 4.

-- What is the takeaway from Figure 2? Based on Figure 2, there is only marginal improvment due to proposed calibration and this figure does not support the claim in abstract that "We demonstrate that our method consistently outperforms prior re-calibration approaches across a diverse set of regression benchmarks and model classes."

-- One could simply apply kernel regression on data and there is no need for modeling and then calibration using kernel regression; the advantage of having a two step process is not well-motivated and demosntrated (see, e.g., https://arxiv.org/pdf/2010.02681).

---

> ### Author Rebuttal · Authors · 2026-03-31
>
> - _Is Widmann et al.'s (2021) method suitable?_
>
> There is a terminology discrepancy in the literature. E.g., in Luo et al.'s, (2022) work
> $\mathbb{P}_{Y|Q} = Q$ is considered as a "Global Calibration", whereas in the regression focused setup (See e.g., Song et al., (2019) Remark 1) the very same condition is considered as conditional calibration.
>
> Our approach explicitly targets auto-calibration and __not__ the stronger notion of local-calibration. Widmann et al.'s (2021) method is designed exactly to test auto-calibration.
>
> - _Contrast with classification results_
>
> We will extend Sec. 2 to contrast our approach with classification results:
>
> --Kull et al., (2017, 2019) achieve post-hoc auto-calibration but rely on parametrizing the calibration map, which is highly restrictive (Song et al., 2019) for continuous regression densities.
>
> --Hebert-Johnson et al., (2018) and Vashistha and Farahi, (2025) focus solely on quantifying local calibration error, not correcting it.
>
> --Luo et al.'s (2022) uses histogram binning of the $[0,1]$ unit interval. Generalizing this discrete approach to condition on elements of the infinite-dimensional $\mathcal{M}(\mathcal{Y})$ is non-trivial and a meaningful contribution.
>
> --Marx et al., (2023) explicitly mention that post-hoc calibration is complementary to their approach. Our non-parametric method fills this exact gap.
>
> We are grateful for the highly relevant references. We are excited to also add a _future work_ section, discussing how local calibration methods from classification are highly relevant to regression, yet currently unexplored.
>
> - _Relation to Conformal Prediction (CP)_
>
> We agree that our approach would be better positioned by discussing in the related works how it parallels / diverges from the CP framework:
>
> Conformal Prediction (CP) is a related framework in that it also utilizes a held-out calibration set to provide / improve reliability guarantees. However, CP fundamentally differs from our approach in its objectives and outputs. Standard CP aims to construct a prediction interval or region that contains the true target with a user-specified marginal probability. While it is practical for robust, worst-case decision-making it does not characterize the probability distribution inside the predicted region. Because it yields sets rather than full predictive distributions, CP cannot be directly used to evaluate probabilistic metrics like expected risk.
>
> Furthermore, the two frameworks operate on different notions of reliability. Standard CP guarantees marginal coverage, and advanced CP methods strive for conditional coverage (conditioned on the feature $X$). In contrast, our proposed framework enforces auto-calibration, which sits structurally between marginal and full feature-conditional calibration.
>
>
> - _Calibration literature, and Guo's study_
>
> We understand the current phrasing implicitly implies that ML has not been focusing on calibration in the past. We are ready to rephrase it, and expand the list of cited works with (Guo et al., 2017):
>
> Calibration has long been a central problem in statistical forecasting (Dawid, 1984) and has been studied for as long as machine learning has existed. Today, it receives constant attention through recent works on post-hoc recalibration (Guo et al., 2017; Song et al., 2019; Gruber & Buettner, 2024).
>
> - $\lambda, \sigma$ _paramters_
>
> Please see response to Reviewer YGbh.
>
> - _Novelty regarding the proposed decomposition_
>
> Please see response to Reviewer mm2D.
>
> - _Purpose of hypothesis testing section and results on Fig. 2_
>
> Assessing the quality of a calibration method __partially__ relies on the hypothesis tests presented in section 3.3.3., which we included in order to make our presentation self-contained. As satisfying calibration is trivial if one doesn't care about sharpness, and measuring sharpness is very challenging (as we don't have the true posterior $\mathbb{P}_{Y|X}$ at hand), we argue that a re-calibrated prediction should be evaluated by testing calibration and studying the mean error score (i.e., fig 1 and fig 2 together).
>
> We are ready to edit the caption of Fig. 2 with a condensed version of this argument.
>
> - _The role of a base model_
>
> We use a base model because learning the full probabilistic mapping $X \mapsto \mathbb{P}\_{Y \mid X}$ directly is difficult. Our method tackles the simpler recalibration step $Q \mapsto \mathbb{P}_{Y | Q}$. This makes our approach complementary to strong predictive models and improves distributional reliability in a second stage.
>
> - No limitations are discussed
>
> Please see response to Reviewer YGbh.

---

> > ### Author Rebuttal · Reviewer_6e8J · 2026-04-02
> >
> > It seems other referees have a positive view of the paper, so I am willing for AC to override this score. Saying that my concerns are not entirely addressed here.
> >
> > > Hebert-Johnson et al., (2018) and Vashistha and Farahi, (2025) focus solely on quantifying local calibration error, not correcting it
> >
> > Not true. Algorithm 1 in Hebert-Johnson et al., (2018) is a learning algorithm not an evaluation algorithm. Vashistha and Farahi (2025) propose KLCE, a method of evaluation; however, KLCE can be used as a regularizer or loss function to achieve calibration; similar to Kumar et al., 2018 and Marx et al., 2024.
> >
> > > Furthermore, the two frameworks operate on different notions of reliability.
> >
> > Why is comparison with CP unnecessary? If the proposed framework is superior (in the setups considered by the authors), why doesn't it show that CP, which operates under different assumptions, fail to achieve the desired performance? Is the goal not to show that existing methods fail in this setup, hence there is a need/gap that the proposed method addresses?
> >
> > > hyper-parameters.
> >
> > Without seeing the results and performing a sensitivity analysis, it is hard to evaluate the statement. The authors described only what they did; no sensitivity analysis or tabular results were presented. Though I agree that now, at least, the results are reproducible and seem not cherry-picked, hence this is a minor concern, and I can agree that the hyperparameters can be studied in more detail in future work.
> >
> > > Base model:
> >
> > What happens if the authors only use kernel regression on the training data compared to the two-step process? Can they present some results on this? To perform calibration properly, one needs to split data into training, validation, calibration training, calibration validation, and test. If everything is done in one step, then we have only training, validation, and test samples. Hence, there is more data for the kernel regression to achieve a better performance (without a base model). Some empirical results can provide justification for the argument here; without that, IMO argument is justified properly.

---

> > > ### Author Response · Authors · 2026-04-07
> > >
> > > - On Classification Literature:
> > >
> > > We apologize for the misplacement of the Hebert-Johnson et al. (2018) reference. This was a citation error during the drafting process.
> > > Our intent was to categorize the method by Hebert-Johnson et al. (2018) alongside Luo et al. (2022), as both approaches fundamentally rely on the binning of confidence values
> > > ($[0,1] \subset \mathbb{R}$).
> > >
> > > - On Calibration Loss as a Regularizer:
> > >
> > > We agree that it is an interesting future research direction to have better control over the calibration-sharpness tradeoff during training of a model.
> > >
> > > However, as Marx et al. (2024) demonstrate, post-hoc recalibration remains essential even when training-time regularization is applied. As our work focuses on the former, we consider our approach complementary to regularization. Consequently, a comparative study of training-time techniques is beyond the scope of this work.
> > >
> > > - Comparison with Conformal Prediction
> > >
> > > We agree that a comparison with Conformal Prediction (CP) strengthens the positioning of our method.
> > >
> > > As standard CP targets constructing prediction sets (intervals) that achieve the prescribed marginal coverage, it is conceptually different to the notion of reliability we considered.
> > >
> > > This raised practical questions about the methodology required for a meaningful comparison.
> > > However, we have performed an evaluation that we believe accurately highlights the key differences between these two approaches.
> > >
> > > After comparing predictive interval coverage, average interval length, and dependence of the PIT transformed observation on the interval, we found that:
> > > 1. Our method can produce predictive intervals that are comparable to the coverage and length to those of [Chernozhukov et al.: Distributional conformal prediction, 2021.](https://arxiv.org/abs/1909.07889)
> > > 2. With our method the dependence between predicted intervals and the PIT transform of the observations tends to be smaller (as the dataset size increases), i.e., the remaining modeling error is more evenly distributed.
> > >
> > > Please see detailed results at:
> > > [CP comparison experiment.](https://anonymous.4open.science/r/recalibration_icml2026-231B/experiment/conformal_prediction/README.md)
> > >
> > > - Base model and two stage modeling
> > >
> > > We appreciate the reviewer’s question, as it raises a valid point regarding data efficiency.
> > > First, we would like to highlight that our original experiments explicitly address the concern regarding data splitting.
> > > The `None(T+C)` baseline evaluates the base models trained on the combined union of the training and calibration datasets.
> > >
> > > We generally found that the base models can sometimes achieve better _scores_ than the two stage modeling, but are oftentimes miscalibrated. This is not surprising in the light of the calibration-sharpness principle presented in section 4.
> > >
> > > To directly answer your question, we conducted a new ablation study to compare our method against standard kernel regression operating on the full training and calibration data.
> > >
> > > The results were aligned with our previous observations. There were significant calibration errors without post-hoc recalibration. Since we used a translation invariant kernel on the input space, the resulting scores were sometimes suboptimal compared to using more flexible ML methods and the two-stage modeling.  Please see the [detailed results here.](https://anonymous.4open.science/r/recalibration_icml2026-231B/experiment/kernel_baseline/README.md)
> > >
> > >
> > > We will add the additional experiments, references, and discussion to the updated version of the submission.

---

### Decision · Program_Chairs · 2026-04-30

**Decision:**

Accept (regular)

**Comment:**

This work introduces a more powerful re-calibration method for non-parametric distribution regression. Prior methods either rely on parametric assumptions, or only guarantees a weak from of calibration known as PIT uniformity. The idea of this work is to map the predictions to an RLHF and estimate the re-calibration mapping in the RLHF, and the proposed method is efficient in terms of running time. The reviewers appreciate the novelty, soundness and significance of the submission. I recommend acceptance.